# A Survey on Large Language Model Acceleration based on KV Cache Management

**Haoyang Li[1]    Yiming Li[2]\*    Anxin Tian[2]\*    Tianhao Tang[2]    Zhanchao Xu[1,3]    Xuejia Chen[1,3]    Nicole Hu[4]    Wei Dong[5]    Qing Li[1]    Lei Chen[2]**

haoyang-comp.li@polyu.edu.hk          {yliix,atian}@connect.ust.hk          ttangae@cse.ust.hk
zhanchaoxu0228@gmail.com          gresham15437@gmail.com          hulan@link.cuhk.edu.hk
wei_dong@ntu.edu.sg    qing-prof.li@polyu.edu.hk    leichen@cse.ust.hk

[1]The Hong Kong Polytechnic University    [2]The Hong Kong University of Science and Technology
[3]Huazhong University of Science and Technology    [4]The Chinese University of Hong Kong [5]Nanyang Technological University

**Reviewed on OpenReview:** `https://openreview.net/pdf?id=z3JZzu9EA3`

## Abstract

Large Language Models (LLMs) have revolutionized a wide range of domains such as natural language processing, computer vision, and multi-modal tasks due to their ability to comprehend context and perform logical reasoning. However, the computational and memory demands of LLMs, particularly during inference, pose significant challenges when scaling them to real-world, long-context, and real-time applications. Key-Value (KV) cache management has emerged as a critical optimization technique for accelerating LLM inference by reducing redundant computations and improving memory utilization. This survey provides a comprehensive overview of KV cache management strategies for LLM acceleration, categorizing them into token-level, model-level, and system-level optimizations. Token-level strategies include KV cache selection, budget allocation, merging, quantization, and low-rank decomposition, while model-level optimizations focus on architectural innovations and attention mechanisms to enhance KV reuse. System-level approaches address memory management, scheduling, and hardware-aware designs to improve efficiency across diverse computing environments. Additionally, the survey provides an overview of both text and multi-modal datasets and benchmarks used to evaluate these strategies. By presenting detailed taxonomies and comparative analyses, this work aims to offer useful insights for researchers and practitioners to support the development of efficient and scalable KV cache management techniques, contributing to the practical deployment of LLMs in real-world applications. The curated paper list for KV cache management is in: https://github.com/TreeAI-Lab/Awesome-KV-Cache-Management.

## 1 Introduction

Large Language Models (LLMs) (Hadi et al., 2023; Zhu et al., 2023), trained on massive corpora, have revolutionized various domains such as natural language processing (Naveed et al., 2023; Min et al., 2024; Xu et al., 2024a), computer vision (Liu et al., 2023a; Zhang et al., 2024c; Berrios et al., 2023), and multi-modal (Zhang et al., 2024a; Cui et al., 2024; Wu et al., 2023) tasks. Their ability to understand context and perform logical reasoning has enabled remarkable success in various fields, such as time series analysis (Jin et al., 2023; Ma et al., 2024a), recommendation (Tan & Jiang, 2023; Wu et al., 2024c), autonomous driving (Yang et al., 2023; Chen et al., 2024b; Fu et al., 2024b), and healthcare (Qiu et al., 2023; Zhou et al., 2023b). These breakthroughs are powered by state-of-the-art architectures and training

---

\*Corresponding Author.

paradigms, enabling models to achieve unparalleled performance across diverse tasks. Prominent LLMs, such as GPT (Brown et al., 2020; Radford et al., 2018; 2019), LLama (Touvron et al., 2023; Dubey et al., 2024), DeepSeek (Dai et al., 2024; DeepSeek-AI et al., 2024; Lu et al., 2024), Mistral (Jiang et al., 2024a), and GLM (Zeng et al., 2023; Du et al., 2022), are built on the foundational transformer architecture (Vaswani et al., 2017), which excels at capturing long-range dependencies in sequential data. However, despite their powerful capabilities, the computational and memory demands of LLMs, particularly during inference, present significant challenges when scaling them to real-world, long-context, and real-time applications.

A critical bottleneck in LLM inference lies in the efficient management of Key-Value (KV) pairs. Recently, caching techniques (Gracioli et al., 2015; Podlipnig & Böszörmenyi, 2003) have been extensively employed to store previously computed intermediate results, allowing their reuse in subsequent inference steps to accelerate the model, such as graph neural networks (Li & Chen, 2021; Li et al., 2023c; Lin et al., 2020). Fortunately, the auto-regressive generation mechanism inherent to LLMs presents an opportunity to leverage KV caching for efficient text generation. Specifically, auto-regressive generation enables LLMs to produce text token by token, with each token conditioned on all previously generated ones. While this approach is highly effective for generating coherent and contextually relevant outputs, it struggles with poor scalability for long input sequences. This limitation arises because LLMs must compute attention values for every pair of tokens, causing the time and space complexity of the attention matrix to grow quadratically with sequence length. To address this issue, the KV cache mechanism stores the key and value matrices from previous decoding steps, allowing them to be reused. This significantly reduces redundant computations, as the model only computes attention values between new tokens and previously processed tokens, avoiding the need to recompute all attention values.

Several recent surveys (Zhu et al., 2023; Zhuang et al., 2023; Park et al., 2024; Wang et al., 2024b; Ding et al., 2023; Miao et al., 2023; Wan et al., 2023; Zhou et al., 2024c; Tang et al., 2024c; Kachris, 2024; Xu et al., 2023; Albalak et al., 2024; Zefan-Cai, 2024) have explored the domain of efficient LLMs. These surveys primarily examine various aspects of LLM efficiency, presenting valuable insights while leaving room for further refinement and innovation. In particular, these works primarily focus on holistic approaches to improving LLM efficiency, examining a wide range of techniques across multiple dimensions, such as data-level optimizations (e.g., prompt engineering), model architecture-level optimizations (e.g., efficient transformer designs), and system-level optimizations (e.g., task scheduling). For instance, Ding et al. (2023) explore efficiency techniques that integrate data-level and model architecture perspectives, while Miao et al. (2023) examine efficient LLM inference from a comprehensive system-level perspective. Similarly, Tang et al. (2024c), Wan et al. (2023), and Xu et al. (2023) provide analyses that encompass data, model, and system-level optimizations, reflecting holistic approaches to LLM acceleration.

On the other hand, some surveys focus on more specialized aspects for LLM acceleration. For example, Zhu et al. (2023), Park et al. (2024), Wang et al. (2024b), and Tang et al. (2024c) focus on model compression as a key aspect of model-level optimization. Similarly, Kachris (2024) examines hardware acceleration strategies tailored for LLMs, while Xu et al. (2023) investigates parameter-efficient tuning approaches. Albalak et al. (2024) discusses data selection strategies to enhance the efficiency of LLM training, and Xia et al. (2024) highlights collaborative techniques, such as speculative decoding (Leviathan et al., 2023; Kim et al., 2024b), to accelerate model inference. Li et al. (2024c) focuses on prompt compression. Similar to our work, Shi et al. (2024), Li et al. (2024a), and Yuan et al. (2024) also explore the use of KV caches to accelerate LLMs. However, our survey is both complementary and more comprehensive, offering a detailed taxonomy of KV cache management for text-based and multi-modal LLMs. We categorize techniques into token-level, model-level, and system-level perspectives and include benchmarks for both text and multi-modal scenarios. In particular, complementing existing KV cache surveys, we provide a detailed comparison of the differences and advantages of existing models at the token-level, model-level, and system-level.

Specifically, this survey provides a comprehensive overview of the current state of KV cache management and its role in accelerating LLM inference. We begin by introducing the transformer architecture and the role of the KV cache in enabling efficient auto-regressive text generation. We then analyze the challenges associated with KV cache management, including its impact on computational complexity, memory usage, and real-time performance. Following this, we present a taxonomy of existing optimization techniques, categorizing them into token-level, model-level, and system-level optimization approaches. Additionally, we

Table 1: Notation Summary

| Symbol | Definition |
|---|---|
| $X$ | Input sequence of tokens |
| $\mathbf{X}$ | Dense representations of $X$ |
| $d_x$ | Dimensionality of the input embeddings. |
| $\mathbf{E}$ | Embedding matrix $\mathbf{E} \in \mathbb{R}^{d_{\text{vocab}} \times d_x}$. |
| $PE(X)$ | Positional encoding |
| $\mathbf{Q}_i, \mathbf{K}_i, \mathbf{V}_i$ | Query, Key, and Value matrices |
| $d_k, d_v$ | Query/Key and Value dimension |
| $\mathbf{W}_{Q_i}, \mathbf{W}_{K_i}, \mathbf{W}_{V_i}$ | Weight matrices for computing $\mathbf{Q}_i, \mathbf{K}_i, \mathbf{V}_i$. |
| $\mathbf{Z}_i$ | Self-attention Output |
| $\mathbf{W}_O$ | Weight matrix |
| $\mathbf{W}_1, \mathbf{W}_2$ | Weight matrices |
| $\mathbf{b}_1, \mathbf{b}_2$ | Bias vectors |
| $t$ | Sequence length index |
| $t_c$ | Number of tokens stored in the KV cache. |
| $\mathbf{K}_i^t, \mathbf{V}_i^t$ | Key and Value at step $t$ |
| $\hat{\mathbf{K}}_i^{t-1}, \hat{\mathbf{V}}_i^{t-1}$ | Cached Key and Value |
| $h$ | Number of attention heads per layer |
| $L$ | Number of transformer layers |
| $P(x_{t+1}|x_1, \cdots, x_t)$ | Conditional probability |

discuss datasets and evaluation metrics used to benchmark these techniques and provide insights into their effectiveness across various tasks and applications.

## 2 Preliminary

Large language models (LLMs), pretrained on vast corpora, have demonstrated superior capabilities in context understanding and logical reasoning. These models have achieved remarkable success across a wide range of tasks in various domains, including natural language processing (Naveed et al., 2023; Min et al., 2024; Xu et al., 2024a) and computer vision (Liu et al., 2023a; Zhang et al., 2024c; Berrios et al., 2023). Mainstream LLMs, such as GPT (Bubeck et al., 2023), Llama (Touvron et al., 2023), and DeepSeek (Dai et al., 2024), are primarily built on the transformer architecture (Vaswani et al., 2017). To explore the role of Key-Value (KV) cache management in accelerating LLM computations, we first outline the core components of the transformer model and then introduce the mechanisms for managing the KV cache to accelerate the LLMs. Important notations in this survey are summarized in Tab. 1.

### 2.1 Transformer Architecture

Transformers (Vaswani et al., 2017) have become the backbone of LLMs due to their ability to efficiently capture long-range dependencies in sequential data, such as text. This capability makes them particularly well-suited for tasks like machine translation, text generation, and image captioning. The transformer architecture follows an encoder-decoder structure, where most LLMs utilize only the decoder component. We first introduce the core components of the Transformer decoder and then describe the critical auto-regressive generation mechanism. Particularly, we do not describe certain components in the transformer, such as normalization, as they do not impact the understanding of KV cache management.

#### 2.1.1 Transformer Decoder

As shown in Fig. 1a, a decoder-based transformer architecture is composed of multiple stacked Transformer blocks, each designed to process sequential data effectively. Typically, a Transformer block consists of two core components, i.e., a Multi-Head Self-Attention (MHSA) mechanism and a Feed Forward Network (FFN). These blocks are arranged sequentially, where the output of one block is passed as input to the next. This

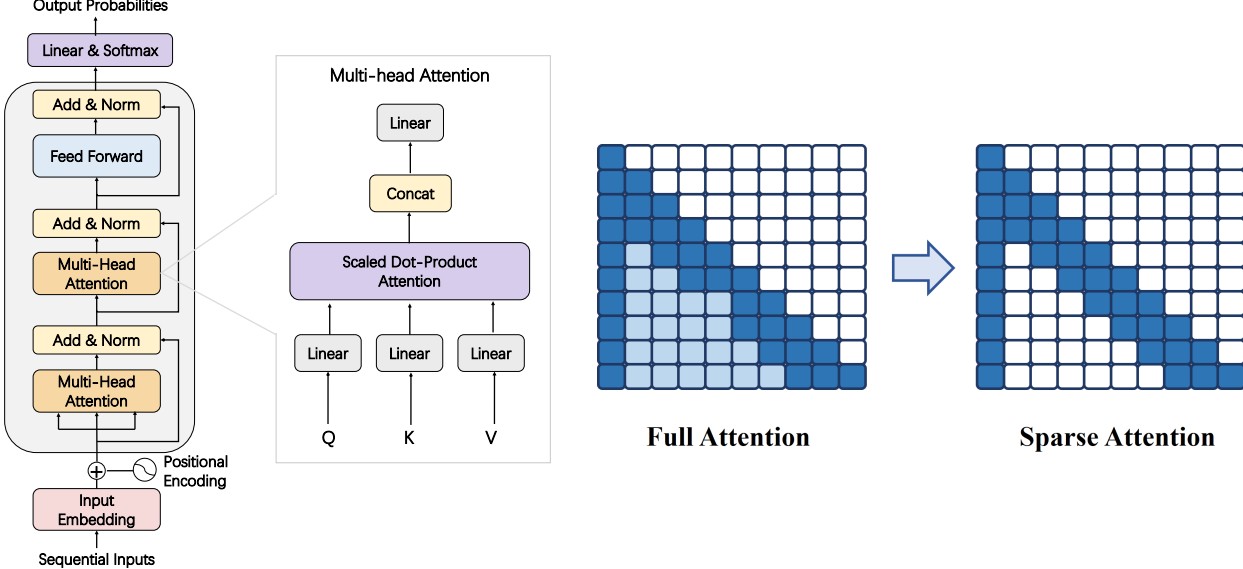

(a) The decoder-only Transformer for LLMs.    (b) The sparsity of attention matrix.

Figure 1: An architecture of transformer and two attention matrices.

iterative design allows the model to refine its understanding of the input sequence progressively, making it highly effective for tasks such as text generation and language modeling.

**Positional Encoding.** Before the input sequence is processed by the Transformer blocks, it undergoes a preprocessing phase. First, a tokenizer processes the input sentence $X$ by splitting it into discrete units, such as words or subwords. The resulting sequence can be represented as $X = [x_1, x_2, \cdots, x_{|X|}]$. These tokens are then mapped to dense vector representations using an embedding layer, i.e., $\mathbf{X} = \mathbf{I}_X \mathbf{E}^\top$, where $\mathbf{I}_X \in \{0,1\}^{n \times d_{\text{vocab}}}$ represents the one-hot vector of tokenized input $X$, $\mathbf{E} \in \mathbb{R}^{d_{\text{vocab}} \times d_x}$ is the embedding matrix, and $\mathbf{X} = [\mathbf{x}_1, \mathbf{x}_2, \cdots, \mathbf{x}_{|X|}] \in \mathbb{R}^{n \times d_x}$ is the resulting matrix of embedded token representations. Since the Transformer architecture does not inherently account for the order of tokens in a sequence, **positional encodings** are added to the token embeddings $\mathbf{X}$ to incorporate positional information. This can be expressed as $\mathbf{X} = \mathbf{X} + PE(X)$, where $PE(X) \in \mathbb{R}^{n \times d_x}$ represents a function (Zhao et al., 2023; Zheng et al., 2021; Su et al., 2024) (e.g., RoPE (Su et al., 2024)) that generates positional embeddings for the input $X$. Note that relative positional embeddings, such as RoPE (Su et al., 2024) (Rotary Positional Embedding), differ significantly from absolute positional encoding. RoPE introduces positional information at each layer of the model through rotational transformations.

**Transformer Block.** Once the input features are prepared, they are passed through a series of stacked Transformer blocks. Each block begins with the Multi-Head Self-Attention (MHSA) mechanism, which captures both local and global dependencies. For each token, the self-attention mechanism computes a weighted sum over all other tokens in the sequence, where the weights are derived from the similarity between the tokens. Particularly, since the operations within each transformer block are identical, we use a single transformer block as an example. Specifically, given the input to a block, denoted as $\mathbf{X} \in \mathbb{R}^{|X| \times d}$, the MHSA mechanism computes the query vectors $\mathbf{Q}_i \in \mathbb{R}^{|X| \times d_k}$, key vectors $\mathbf{K}_i \in \mathbb{R}^{|X| \times d_k}$, and value vectors $\mathbf{V}_i \in \mathbb{R}^{|X| \times d_v}$. These vectors are obtained through learned linear transformations as follows:

$$\mathbf{Q}_i = \mathbf{X}\mathbf{W}_{Q_i}, \quad \mathbf{K}_i = \mathbf{X}\mathbf{W}_{K_i}, \quad \mathbf{V}_i = \mathbf{X}\mathbf{W}_{V_i}, \tag{1}$$

where $\mathbf{W}_{Q_i} \in \mathbb{R}^{d_x \times d_k}$, $\mathbf{W}_{K_i} \in \mathbb{R}^{d_x \times d_k}$ and $\mathbf{W}_{V_i} \in \mathbb{R}^{d_x \times d_v}$ are the learned weight parameters. Then, the self-attention operation is applied to each triple $(\mathbf{Q}_i, \mathbf{K}_i, \mathbf{V}_i)$, and obtains the output of the $i$-th attention head $\mathbf{Z}_i$ as follows:

$$\mathbf{Z}_i = \text{Attention}(\mathbf{Q}_i, \mathbf{K}_i, \mathbf{V}_i) = \text{Softmax}\left(\frac{\mathbf{Q}_i \mathbf{K}_i^\top}{\sqrt{d_k}}\right) \mathbf{V}_i, \tag{2}$$

where $\sqrt{d_k}$ is a scaling factor to ensure numerical stability. To capture diverse relationships, multiple attention heads with $h$ heads are applied to $\mathbf{X}$ in parallel, and their outputs are concatenated with one transformation as follows:

$$\mathbf{Z} = \text{Concat}(\mathbf{Z}_1, \mathbf{Z}_2, \ldots, \mathbf{Z}_h)\mathbf{W}_O, \tag{3}$$

where Concat is the concatenation operation and $\mathbf{W}_O \in \mathbb{R}^{d_v \times d_o}$ are the trainable parameters.

Following the self-attention mechanism, the output is passed through a **Feed Forward Network (FFN)**. The FFN is a fully connected neural network that applies two linear transformations separated by a nonlinear activation function $\sigma(\cdot)$ (e.g., ReLU (Agarap, 2018)) :

$$\text{FFN}(\mathbf{Z}) = \sigma(\mathbf{Z}\mathbf{W}_1 + \mathbf{b}_1)\mathbf{W}_2 + \mathbf{b}_2 \tag{4}$$

where $\mathbf{W}_1 \in \mathbb{R}^{d_o \times d_1}$ and $\mathbf{W}_2 \in \mathbb{R}^{d_1 \times d_2}$ are two parameters, $\mathbf{b}_1 \in \mathbb{R}^{d_1}$ and $\mathbf{b}_2 \in \mathbb{R}^{d_2}$ are two bias vectors.

### 2.1.2 Auto-regressive Generation Mechanism

LLMs employ an autoregressive mechanism to generate text token by token, with each token conditioned on the previously generated ones. This iterative process ensures that the output sequence remains coherent and contextually appropriate. Formally, given an input sequence of tokens $X = [x_1, x_2, \cdots, x_t]$, the model predicts the next token $x_{t+1}$ at each decoding step $t$ by modeling the conditional probability distribution as follows:

$$P(x_{t+1}|x_1, x_2, \cdots, x_t) = \text{Softmax}(\mathbf{h}_t\mathbf{W}_{\text{out}} + \mathbf{b}_{\text{out}}), \tag{5}$$

where $\mathbf{h}_t \in \mathbb{R}^{d_h}$ represents the hidden state of the LLM regarding $X$ at step $t$, $\mathbf{W}_{\text{out}} \in \mathbb{R}^{d_h \times vocab}$ is the output projection matrix, and $\mathbf{b}_{\text{out}}$ is the bias vector. The softmax function converts the logits into a probability distribution over the vocabulary. Then, at each decoding step, the model generates the next token $x_{t+1}$ by sampling from the predicted probability distribution:

$$x_{t+1} \sim P(x_{t+1}|x_1, x_2, \cdots, x_t). \tag{6}$$

The generated token $x_{t+1}$ is then appended to the sequence $X = [x_1, \cdots, x_t, x_{t+1}]$, and the process continues until a special end-of-sequence (EOS) token is generated or a predefined maximum length is reached.

## 2.2 Key-Value Cache in Transformer Models

Auto-regressive generation is a powerful mechanism that enables LLMs to produce high-quality, contextually coherent text. However, it presents computational challenges for long sequences, as the Keys and Values need to be recomputed for each token during the generation process. The KV cache optimization addresses this issue by storing the previously computed Keys and Values and reusing them for subsequent token generation, thereby reducing redundant computations and improving inference efficiency.

### 2.2.1 Auto-regressive Generation with KV Cache

Here, we describe how caching KV pairs of tokens accelerates LLM inference. Specifically, at each decoding step $t$, the model performs self-attention over the entire sequence $X = [x_1, \cdots, x_{t-1}, x_t]$ to generate the next token $x_{t+1}$. This process requires the computation of Keys and Values matrices for all previously processed tokens in $X = [x_1, \cdots, x_t]$. Notably, when generating the token $x_t$, the LLM has already computed the Keys and Values for the tokens in $X[1 : t-1] = [x_1, \cdots, x_{t-1}]$. The KV cache optimizes this process by storing the previously computed Keys and Values matrices for $X[1 : t-1]$ and reusing them, thereby only requiring the computation of Keys and Values for the new token $x_t$. This significantly improves efficiency by eliminating redundant computations.

Formally, at each decoding step $t$, the new token embedding $\mathbf{x}_t$ is used to compute the query vector $\mathbf{q}_i^t$, key vector $\mathbf{k}_i^t$, and value vector $\mathbf{v}_i^t$ as follows:

$$\mathbf{q}_i^t = \mathbf{x}_t\mathbf{W}_{Q_i}, \quad \mathbf{k}_i^t = \mathbf{x}_t\mathbf{W}_{K_i}, \quad \mathbf{v}_i^t = \mathbf{x}_t\mathbf{W}_{V_i}, \tag{7}$$

The newly computed $\mathbf{k}_i^t$ and $\mathbf{v}_i^t$ are then appended to the cached key and value matrices from previous steps:

$$\mathbf{K}_i^t = \text{Concat}(\hat{\mathbf{K}}_i^{t-1}, \mathbf{k}_i^t), \ \mathbf{V}_i^t = \text{Concat}(\hat{\mathbf{V}}_i^{t-1}, \mathbf{V}_i^t), \tag{8}$$

where $\hat{\mathbf{K}}_i^{t-1} \in \mathbb{R}^{t-1 \times d_k}$ and $\hat{\mathbf{V}}_i^{t-1} \in \mathbb{R}^{t-1 \times d_v}$ represent the cached key and value matrices of tokens in $X[1:t-1]$. These cached matrices are then used in the scaled dot-product attention computation for the token $x_t$. The attention output $\mathbf{z}_i^t$ for the token $x_t$ at step $t$ is calculated as:

$$\mathbf{z}_i^t = \text{Softmax}\left(\frac{\mathbf{q}_i^t \mathbf{K}_i^{t\top}}{\sqrt{d_k}}\right) \mathbf{V}_i^t, \tag{9}$$

Then, a similar KV reuse process can be applied to different attention heads in each layer of the LLM.

### 2.2.2 Time and Space Complexity Analysis

Given a transformer-based $L$-layer LLM with $h$ attention heads per layer and an input sequence of length $X = [x_1, \cdots, x_t]$, we analyze the time saved and the space required to store cached KV pairs. For simplicity, we assume that the keys and values of the $t_c$ tokens are stored for all heads across all LLM layers.

**Computational Complexity.** For each token, the saved computation time comes from avoiding the repeated computation of Keys and Values in Equation 1, the self-attention result in Equation 2, and the linear transformation in Equation 3. We omit the time analysis on operations in transformer that do not affect the understanding of KV cache acceleration, such as layer norm and position encoding.

- **QKV Computation.** The time of computing Queries, Keys and Values for each token in Equation 1 is $\triangle_1 = O(2d_x d_k + d_x d_v)$.

- **Self-attention Result.** The time complexity of computing each attention score $\mathbf{z}_i$ in Equation 2 takes $O(t(d_k + d_v))$.

- **Linear Transformation.** To merge the $h$ attention results in Equation 3 the time is $\triangle_2 = O(hd_v + d_v d_o)$.

Therefore, for $t_c$ cached tokens across $h$ attention heads and $L$ layers, the total saved computation time is:

$$O\left(L \cdot h \cdot t_c \cdot t \cdot (d_k + d_v) + L \cdot h \cdot t_c \left(\triangle_1 + \triangle_2\right)\right) \tag{10}$$

Thus, the saved time is directly proportional to the number of cached tokens $t_c$, significantly accelerating model computation, especially for longer sequences (when $t$ is large).

**Space Complexity.** Compared to computation without caching, additional space is required to store the cached KV pairs for $t_c$ tokens across $h$ attention heads and $L$ layers. Assuming each Key and Value is stored in Float16 precision, the total extra space needed can be expressed as:

$$O(L \cdot h \cdot t_c \cdot (d_k + d_v) \cdot sizeof(Float16)) \tag{11}$$

Thus, for the same LLM model, the extra space required to store the KV pairs primarily depends on the number of cached tokens and the precision of the cached Keys and Values. To address this, existing approaches explore various techniques to reduce the extra space consumption, such as caching only the most important Keys and Values or applying quantization techniques to lower the bit precision of the stored Keys and Values.

### 2.3 Challenges in KV Cache Management

As analyzed in Sec. 2.2.2, reusing cached KV pairs enables the LLM to avoid recomputing past tokens, resulting in significant speedups during inference. However, as sequence lengths grow, the size of the KV cache increases proportionally, placing significant pressure on memory. Consequently, it becomes challenging to manage this cache effectively to accelerate LLM computation without excessive space usage.

- **Cache Eviction Policies:** Determining which items to evict when the cache reaches its capacity is a complex problem. Popular policies (Podlipnig & Böszörmenyi, 2003) like Least Recently Used (LRU) or Least Frequently Used (LFU) do not align with LLM patterns, leading to suboptimal performance.

- **Memory Management:** The memory required for the KV cache grows linearly with both the sequence length and the number of layers, which can quickly exceed the hardware memory limits, especially for long sequences. Consequently, managing the collaboration between different types of storage hardware (e.g., GPU, CPU, or external memory) becomes a significant challenge.

- **Latency Bottlenecks:** Accessing and updating the cache at each decoding step can introduce latency, particularly for hardware with limited memory bandwidth.

- **Compression Trade-offs:** Compressing the KV cache can reduce memory usage but may degrade model performance if key information is lost.

- **Dynamic Workloads:** Handling dynamic and unpredictable workloads, where access patterns and data requirements frequently change, requires adaptive caching strategies that can respond in real time.

- **Distributed Coordination:** In distributed KV caches, maintaining coordination across multiple nodes to ensure consistency, fault tolerance, and efficient resource usage adds significant complexity.

## 3    Taxonomy

In the above sections, we analyzed how the number of cached Key-Value (KV) pairs significantly impacts both the computation time and the additional memory required during inference. Efficient KV cache management is critical to balancing performance improvements and resource utilization, especially as sequence lengths and model sizes continue to grow. After carefully reviewing existing approaches, we categorize KV cache optimization strategies into three levels: token-level optimization, model-level optimization, and system-level optimization. Each level addresses specific aspects of the challenges associated with KV cache management and offers distinct techniques to enhance efficiency. The detailed taxonomy is illustrated in Fig. 2.

- **Token-Level Optimization** refers to improving KV cache management efficiency by focusing on the fine-grained selection, organization, and compression at the token level, requiring no architectural changes to the original model. While KV cache selection (Sec. 4.1) focuses on prioritizing and storing only the most relevant tokens, KV cache budget allocation (Sec. 4.2) dynamically distributes memory resources across tokens to ensure efficient cache utilization under limited memory. Furthermore, KV cache merging (Sec. 4.3) reduces redundancy by combining similar or overlapping KV pairs, while KV Cache Quantization (Sec. 4.4) minimizes the memory footprint by reducing the precision of cached KV pairs. Finally, KV cache low-rank decomposition (Sec. 4.5) uses low-rank decomposition techniques to reduce cache size.

- **Model-level Optimization** refers to designing an efficient model structure to optimize KV cache management. This can further refer to several strategies: Attention grouping and sharing (Sec. 5.1) methods examine the redundant functionality of keys and values and group and share KV cache within or across transformer layers. Architecture alterations (Sec. 5.2) emerge to design new attention mechanisms or construct extrinsic modules for KV optimization. Furthermore, there are also works designing or combining non-transformer architectures (Sec. 5.3) that adopt other memory-efficient designs like recurrent neural networks to optimize the KV cache in traditional transformers.

- **System-level Optimization** refers to optimizing the KV Cache management through two classic low-level aspects: memory management (Sec. 6.1) and scheduling (Sec. 6.2). While memory management techniques focusing on architectural innovations like virtual memory adaptation, intelligent prefix sharing, and layer-aware resource allocation, scheduling strategies have evolved to address diverse optimization goals through prefix-aware methods for maximizing cache reuse, preemptive techniques for fair context switching, and layer-specific mechanisms for fine-grained cache control. In addition, we provide a detailed introduction for hardware accelerator design in Sec. 6.3, including single/multi-GPU, I/O-based solutions, heterogeneous computing and SSD-based solutions.

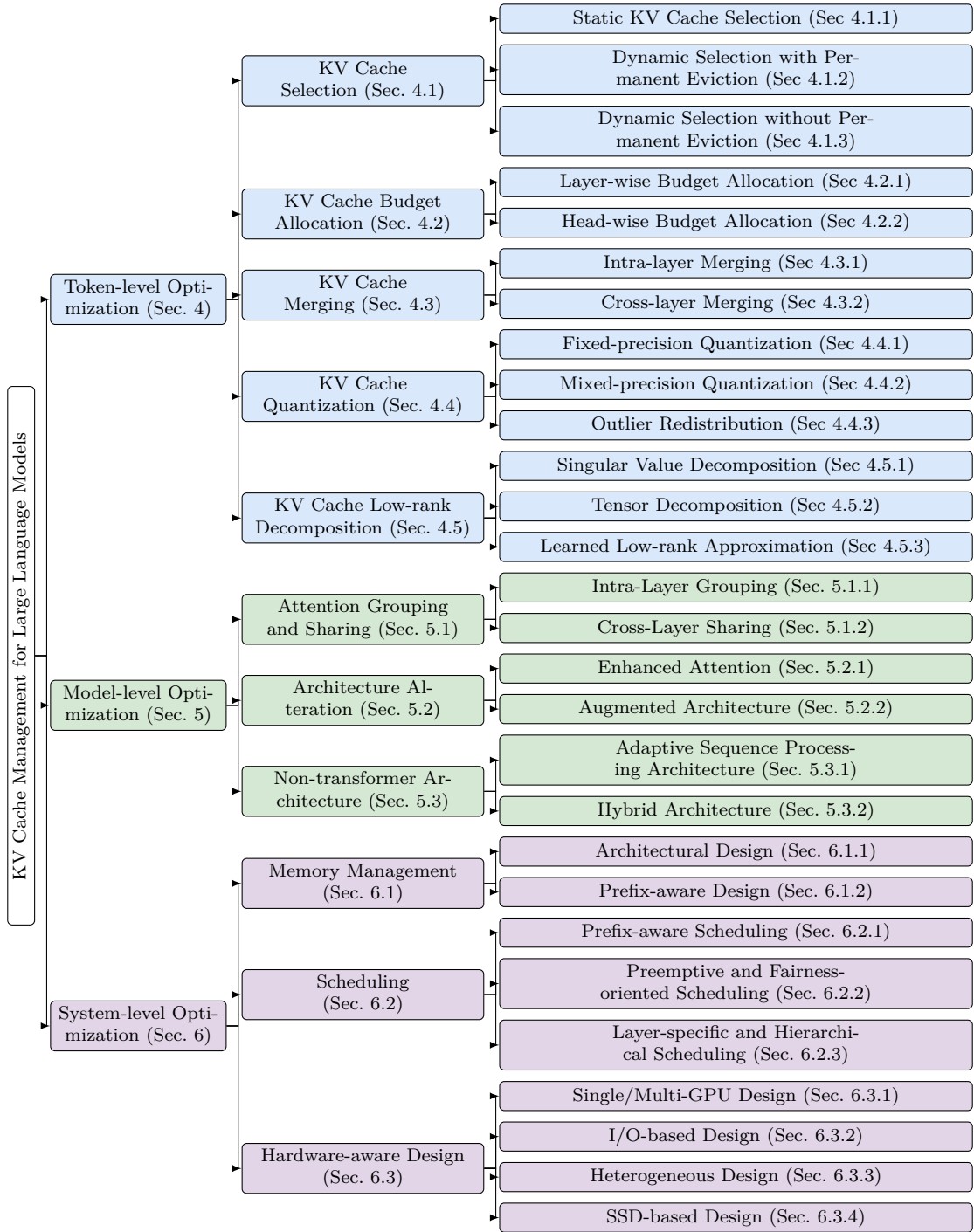

Figure 2: Taxonomy of KV Cache Management for Large Language Models.

## 4 Token-level Optimization

At the token level, optimization focuses exclusively on improving the KV cache management based on the characteristics and patterns of the KV pairs of tokens, without considering enhancements from model architecture improvements or system parallelization techniques. Generally, token-level optimization methods are guided by observations from LLMs and sequential inputs. Existing approaches can be categorized into

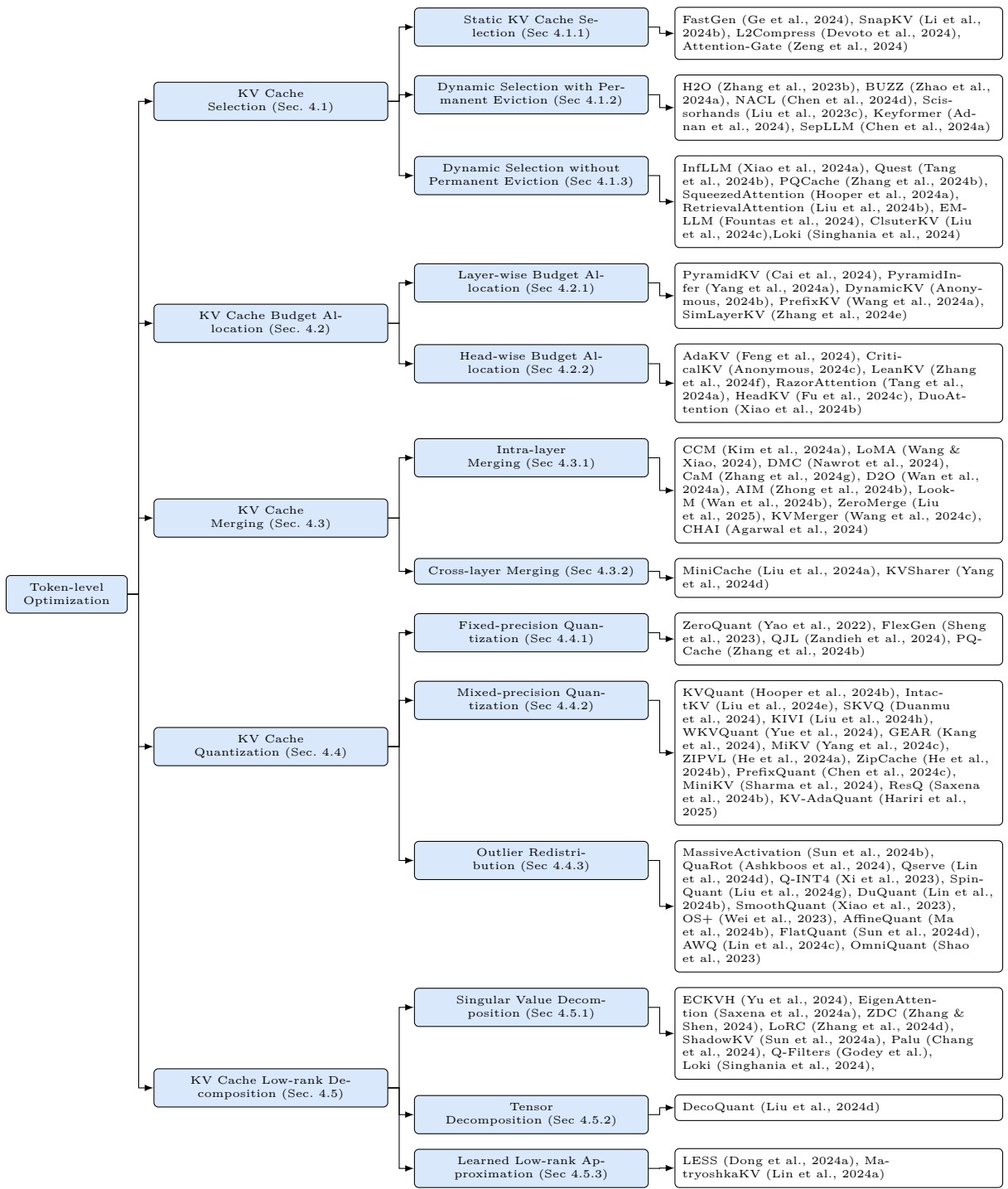

Figure 3: Taxonomy of the Token-level Optimization for KV Cache Management.

five main types: KV cache selection, KV cache budget allocation, KV cache merging, KV cache quantization, and KV cache low-rank decomposition. The taxonomy of token-level optimization is shown in Fig. 3.

## 4.1 KV Cache Selection

Table 2: Comparison of KV cache selection strategies.

| Method | Initial tokens | Top-$k$ tokens | Recent tokens | Permanent eviction | Dynamic selection | Selection granularity | Remark |
|---|---|---|---|---|---|---|---|
| **FastGen** (Ge et al., 2024) | ✓ | ✓ | ✓ | ✓ | | token | five attention structures |
| **SnapKV** (Li et al., 2024b) | | ✓ | ✓ | ✓ | | token | observation window-based |
| **L2Compress** (Devoto et al., 2024) | | ✓ | | | | token | $L_2$ norm-based importance |
| **Attention-Gate** (Zeng et al., 2024) | | ✓ | | ✓ | | token | learned eviction policy |
| **StreamingLLM** (Xiao et al., 2024c) | ✓ | | ✓ | ✓ | ✓ | token | initial and recent tokens |
| **LM-Infinite** (Han et al., 2024) | ✓ | | ✓ | ✓ | ✓ | token | distance ceiling |
| **H2O** (Zhang et al., 2023b) | | ✓ | ✓ | ✓ | ✓ | token | accmulative attention score |
| **BUZZ** (Zhao et al., 2024a) | ✓ | ✓ | ✓ | ✓ | ✓ | token | beehive-like structure |
| **Scissorhands** (Liu et al., 2023c) | | ✓ | ✓ | ✓ | ✓ | token | persistence of importance |
| **NACL** (Chen et al., 2024d) | | ✓ | ✓ | ✓ | ✓ | token | diversified random eviction |
| **Keyformer** (Adnan et al., 2024) | | ✓ | ✓ | ✓ | ✓ | token | gumbel logit adjustment |
| **InfLLM** (Xiao et al., 2024a) | ✓ | ✓ | ✓ | | ✓ | block | block-level KV management |
| **Quest** (Tang et al., 2024b) | | ✓ | | | ✓ | block | new block representation |
| **PQCache** (Zhang et al., 2024b) | ✓ | ✓ | ✓ | | ✓ | block | product quantization |
| **SqueezedAttention** (Hooper et al., 2024a) | | ✓ | | | ✓ | cluster | hierarchical clusters |
| **RetrievalAttention** (Liu et al., 2024b) | ✓ | ✓ | ✓ | | ✓ | token | ANN search |
| **EM-LLM** (Fountas et al., 2024) | ✓ | ✓ | ✓ | | ✓ | event | episodic events |
| **SparQ** (Ribar et al., 2024) | | ✓ | ✓ | | ✓ | token | low-dimensional retrieval |
| **InfiniGen** (Lee et al., 2024) | | ✓ | | | ✓ | token | asynchronous prefetching |
| **RecycledAttention** (Xu et al., 2024b) | | ✓ | ✓ | | ✓ | token | periodic top-$k$ selection |
| **MagicPIG** (Chen et al., 2024g) | ✓ | ✓ | ✓ | | ✓ | token | Local Sensitive Hash |

As shown in Fig. 1b, the attention matrix is sparse. KV cache selection mechanisms have emerged as a critical optimization strategy, aimed at reducing memory utilization of KV caches, minimizing inference latency, and enhancing overall throughput in large language models. These optimization objectives have driven the development of various selection methodologies, which can be classified into two distinct categories: (1) **static KV cache selection**, which performs token filtering exclusively during the prefilling phase, with selected tokens remaining fixed throughout subsequent decoding steps; and (2) **dynamic KV cache selection**, which continuously updates KV cache during the decoding phase, enabling adaptive cache management. In dynamic KV cache selection approaches, KV cache tokens that are not selected may be permanently evicted or offloaded to hierarchical caching devices such as CPU memory, implementing a multi-tier storage strategy. Given that real-time KV cache selection during decoding may incur substantial computational overhead, several studies have focused on developing optimized retrieval algorithms to enhance the efficiency of this process. These optimizations include block-level retrieval instead of token-level granularity to reduce search

complexity, asynchronous query mechanisms to hide latency, and parallel retrieval pipelines to accelerate the selection process. These optimization efforts aim to mitigate the computational burden while maintaining the effectiveness of token selection. The summary of the KV cache selection is listed in Tab. 2.

### 4.1.1 Static KV Cache Selection

Static KV cache selection methods perform a one-time compression on the KV Cache immediately after the prefilling phase is completed. The model then uses this compressed KV cache for subsequent decoding inference. FastGen (Ge et al., 2024) introduces a pattern-aware approach by identifying five fundamental attention structures and implementing targeted selection strategies. These include proximity-based retention for local attention patterns, selective preservation of critical tokens for punctuation-focused attention, frequency-based filtering for sparse attention distributions, and complete token retention for broad attention patterns. SnapKV (Li et al., 2024b) simplifies FastGen's approach by focusing solely on retrieving tokens based on their importance scores. It demonstrates that among all prompt tokens, only a portion carries crucial information for response generation, with these tokens maintaining their significance during the generation phase. The approach employs an end-positioned observation window to detect these important contextual tokens. Their corresponding key-value pairs are then concatenated with the tokens from the observation window. L2Compress (Devoto et al., 2024) proposes another simple yet effective approach to identify important tokens during the prefilling phase using the $L_2$ norm for key embeddings. By analyzing attention distributions in decoder-only Transformers, the authors find a clear correlation between the $L_2$ norm of a key embedding and its attention score during decoding, where a low $L_2$ norm usually leads to high attention scores. Based on this observation, they propose a strategy that compresses the KV Cache by retaining tokens with lower $L_2$ norms. Attention-Gate (Zeng et al., 2024) introduces a learnable KV-Cache eviction mechanism that processes the entire context sequence and generates token-wise eviction decisions through a parameterized policy network, enabling dynamic in-context memory management.

### 4.1.2 Dynamic Selection with Permanent Eviction

This category of methods performs frequent KV cache selection during the decoding phase, permanently removing unselected KV cache tokens from memory. Early works employ a sliding-window mechanism to address long-text inference challenges, where tokens falling outside the window are permanently evicted and become inaccessible. StreamingLLM (Xiao et al., 2024c) uncovers a crucial phenomenon in transformer attention where preserved key-value pairs from initial sequence tokens maintain crucial model performance. This attention sink effect manifests through asymmetric attention weight accumulation at early positions, regardless of semantic significance. The approach leverages this characteristic by incorporating attention sink positions with recent context for efficient processing. LM-Infinite (Han et al., 2024) demonstrates that conventional techniques, including sliding-window patterns and relative positional encodings, fail to resolve length generalization issues. The study introduces a novel methodology through the integration of Λ-shaped attention masking and attention distance ceiling mechanisms.

Recent works have explored leveraging attention scores as a criterion for selecting significant KV cache tokens. H2O (Zhang et al., 2023b) observes that attention computations are primarily driven by a select group of high-impact tokens, known as Heavy Hitters. This method reformulates cache optimization as a dynamic submodular problem, utilizing cumulative attention scores to guide token retention decisions. Unlike H2O, BUZZ (Zhao et al., 2024a) employs a beehive-like structure that selects Heavy Hitters in local KV cache segments. NACL (Chen et al., 2024d) identifies a fundamental limitation in H2O, namely their dependence on potentially biased local attention statistics. To overcome this issue, they develop an alternative approach implementing a diversified random eviction strategy for token selection. Scissorhands (Liu et al., 2023c) builds upon the temporal significance principle, which suggests that tokens demonstrating historical importance maintain their influence in subsequent computational steps. This observation enables the preservation of repetitive attention patterns through selective token retention. Additionally, Keyformer (Adnan et al., 2024) reveals that token removal distorts the underlying softmax probability distribution. Considering the pivotal role of softmax distributions in token significance evaluation, they incorporate regularization techniques to mitigate these distributional perturbations. SepLLM (Chen et al., 2024a) observes that separator tokens (e.g., commas, periods, and line breaks) receive disproportionately high attention scores and naturally summarize

text segments. Building on this, SepLLM retains separator tokens together with initial tokens, important tokens, and recent tokens in the cache.

### 4.1.3 Dynamic Selection without Permanent Eviction

The aforementioned permanent eviction-based approaches face two significant limitations. First, the irreversible eviction of tokens potentially impairs the model's performance on long-sequence tasks, particularly in needle-in-a-haystack scenarios, and these methods prove challenging to adapt to multi-turn dialogue contexts. Second, KV cache selection during the decoding phase introduces computational overhead, adversely affecting decoding latency and compromising end-to-end acceleration. To address these challenges, several studies have focused on developing decoding-phase KV cache selection strategies without permanent eviction. These approaches typically employ multi-tier cache systems (e.g., CPU-GPU hierarchical caching) and leverage advanced data structures and system-level enhancements to optimize retrieval efficiency, enabling efficient inference with reduced GPU KV cache footprint.

To accelerate the retrieval of critical tokens, several research efforts have proposed index-based approaches that organize and access KV cache at block or cluster granularity, enabling efficient query and extraction operations. InfLLM (Xiao et al., 2024a) maintains full KV cache in blocks while facilitating long sequence processing through a hierarchical storage strategy. The framework employs CPU-GPU memory orchestration, preserving essential tokens and current computational units in GPU memory while offloading less frequently accessed units to CPU memory. To further enhance top-$k$ block retrieval precision, the Quest (Tang et al., 2024b) framework presents a refined block representation approach based on minimal and maximal key values in KV cache blocks. PQCache (Zhang et al., 2024b) also implements block-based KV cache management and identifies salient tokens through Maximum Inner-Product Search (MIPS), leveraging Product Quantization (PQ) codes and centroids. SqueezedAttention (Hooper et al., 2024a) employs K-means clustering in an offline stage to group semantically similar keys, with each group represented by a centroid. During inference, it compares input queries against these centroids to identify and load only the semantically relevant keys from the context. Similarly, RetrievalAttention (Liu et al., 2024b) manages KV cache tokens using approximate nearest neighbor search (ANNS) techniques. Additionally, EM-LLM (Fountas et al., 2024) dynamically segments incoming tokens into episodic events. Also, it implements a hybrid retrieval mechanism that combines semantic similarity matching with temporal context to efficiently access relevant KV cache segments. Similarly, ClusterKV (Liu et al., 2024c) groups tokens into semantic clusters and selectively recalls them during inference, achieving both high accuracy and efficiency for LLMs.

To accelerate top-$k$ token identification, SparQ (Ribar et al., 2024) identifies the $r$ most significant elements in the incoming query vector and selectively retrieves the corresponding components along the hidden dimension of the cached key matrix $K$ for approximate attention computation. To overlap prefetching latency, InfiniGen (Lee et al., 2024) employs asynchronous prefetching, utilizing indices of salient KV entries selected by queries from the previous layer to retrieve KV cache entries in the current layer. To ensure maximum model performance, RecycledAttention (Xu et al., 2024b) sustains the entire KV cache during inference computations, yielding no improvements in memory efficiency. The approach performs periodic top-$k$ token selection to identify salient tokens. Moreover, MagicPIG (Chen et al., 2024g) shows that attention-based top-$k$ selection may incur performance degradation. To address this limitation, they introduce a novel heterogeneous computing framework leveraging Locality Sensitive Hashing (LSH) techniques. The system stores LSH hash tables and performs attention estimation on CPU. Recently, Loki (Singhania et al., 2024) is a sparse attention method motivated by the observation that attention key vectors lie in a low-dimensional space, leveraging PCA-based dimensionality reduction and dynamic top-k token selection to significantly reduce computation and memory overhead while preserving model accuracy.

### 4.1.4 Summary and Future Directions

Static KV cache selection algorithms demonstrate superior decoding efficiency overall; however, their efficacy remains to be thoroughly validated in multi-turn dialogues and extended decoding length scenarios. Dynamic KV cache selection algorithms, while adaptive, introduce additional computational overhead during the decoding phase due to frequent cache selection operations. Multi-tier cache architectures and prefetching schemes partially mitigate these challenges, yet their capability to achieve rapid and accurate

Table 3: Comparison of KV cache budget allocation strategies. **Extra**: Extra-calibration

| Method | Layer-wise | Head-wise | Retrieval-head | Input-specific | Extra | Remark |
|---|---|---|---|---|---|---|
| **PyramidKV** (Cai et al., 2024) | ✓ | | | | | pyramid-shaped |
| **PyramidInfer** (Yang et al., 2024a) | ✓ | | | | | pyramid-shaped |
| **DynamicKV** (Anonymous, 2024b) | ✓ | | | ✓ | | maximize attention retention rate |
| **PrefixKV** (Wang et al., 2024a) | ✓ | | | ✓ | | maximize attention retention rate |
| **CAKE** (Anonymous, 2024a) | ✓ | | | ✓ | | layer-specific preference score |
| **SimLayerKV** (Zhang et al., 2024e) | ✓ | | | ✓ | | KV cache compression for lazy layers |
| **AdaKV** (Feng et al., 2024) | | ✓ | | ✓ | | minimize attention computation loss |
| **CriticalKV** (Anonymous, 2024c) | | ✓ | | ✓ | | minimize attention computation loss |
| **LeanKV** (Zhang et al., 2024f) | | ✓ | | ✓ | | maximize attention retention rate |
| **RazorAttention** (Tang et al., 2024a) | | ✓ | ✓ | | ✓ | echo and induction heads |
| **HeadKV** (Fu et al., 2024c) | | ✓ | ✓ | | ✓ | retrieval and reasoning heads |
| **DuoAttention** (Xiao et al., 2024b) | | ✓ | ✓ | | ✓ | learned retrieval heads |

retrieval within acceptable decoding latency constraints requires further empirical validation, particularly in real-world applications involving long sequences. Furthermore, existing selection methods predominantly rely on attention score-based top-$k$ selection mechanisms. However, current top-$k$ approaches may not be able to effectively identify and extract relevant tokens in ultra-long sequence tasks.

## 4.2 KV Cache Budget Allocation

The hierarchical architecture of LLMs leads to diverse information extraction patterns across layers, with each layer's KV-cache contributing differently to model performance. This inherent heterogeneity indicates that uniform KV-cache compression across layers may be suboptimal. KV cache budget allocation addresses this challenge by intelligently distributing memory resources based on each component's importance to prediction accuracy, thereby optimizing memory utilization while minimizing accuracy degradation. It is worth noting that budget allocation approaches prioritize the effective allocation of computational budget rather than token selection strategies. Current budget allocation strategies can be categorized into two levels of granularity: **layer-wise** budget allocation, which assigns different compression ratios across model layers, and the more fine-grained **head-wise** budget allocation, which enables precise memory distribution across individual attention heads within each layer, offering more flexible and targeted optimization opportunities. The summary of KV budget allocation is listed in Tab. 3.

### 4.2.1 Layer-wise Budget Allocation

In contrast to conventional approaches with uniform KV cache sizes, PyramidKV (Cai et al., 2024) employs a pyramid-shaped memory allocation strategy, assigning larger cache capacities to lower layers that progressively decrease in upper layers. This design is supported by the observation that lower layers exhibit uniform attention distributions across input sequences, while upper layers show concentrated attention on specific tokens. PyramidInfer (Yang et al., 2024a) also adopts a pyramid-shaped budget allocation strategy

while selecting tokens with high attention values at each layer. Additionally, during the decoding phase, PyramidInfer dynamically maintains a set of significant tokens through frequent updates driven by attention values. Unlike previous methods, DynamicKV (Anonymous, 2024b) implements an input-adaptive budget allocation strategy by analyzing attention patterns. Specifically, it computes the average attention scores between recent and historical tokens, identifies the top-$k$ tokens with the highest attention values across layers, and proportionally distributes the budget based on the density of significant tokens in each layer. Similarly, PrefixKV (Wang et al., 2024a) identifies the most important tokens for each layer by computing the average attention score of tokens within that layer. PrefixKV (Wang et al., 2024a) then uses a unified threshold to determine the number of retained tokens, adaptively adjusting the retention for each layer based on its importance distribution. CAKE (Anonymous, 2024a) examines attention scores through two lenses: the spatial distribution of inter-token attention and the temporal evolution of attention focus. These measurements are combined to compute layer-specific importance scores, which further guide the allocation of memory resources. Additionally, SimLayerKV (Zhang et al., 2024e) identifies lazy layers, which are ineffective at capturing long-range dependencies. The framework then selectively preserves cache entries, maintaining only the initial and recent tokens for lazy layers, while retaining the complete KV cache for non-lazy layers.

### 4.2.2   Head-wise Budget Allocation

AdaKV (Feng et al., 2024) leverages the observation that attention patterns exhibit distinct concentrations across different heads. It implements head-specific memory allocation by optimizing an L1 loss bound between the original and pruned multi-head attention outputs. Within the constraints of a layer-wise budget, the method distributes cache capacity among heads to maximize the preserved attention information collectively. Building upon AdaKV, CriticalKV (Anonymous, 2024c) introduces significant enhancements by recognizing that the importance of KV cache entries extends beyond attention weights to encompass value states and pretrained parameter matrices. Leveraging this insight, the framework implements a novel selection algorithm that identifies essential cache entries by minimizing the maximum potential output perturbation. LeanKV (Zhang et al., 2024f) implements a fine-grained memory optimization strategy that operates independently for each attention head and input request. The method identifies the smallest subset of tokens necessary to preserve the majority of information flow, allocating cache space based on a predefined attention score threshold and typically maintaining 95% of the total attention mass.

Retrieval head-based methods represent a specialized category of head-wise allocation strategies that focus on identifying and prioritizing attention heads crucial for extracting key information from long sequences. This approach allocates larger cache budgets to these specialized heads, known as retrieval heads (Wu et al., 2024d), due to their significant role in information extraction. RazorAttention (Tang et al., 2024a) characterizes two distinct categories of retrieval heads: echo heads, which focus on previously occurring identical tokens, and induction heads, which attend to antecedent tokens that precede current token repetitions. This framework implements differential caching strategies, maintaining complete cache entries for retrieval heads while condensing remote tokens into consolidated compensation tokens for non-retrieval heads. HeadKV (Fu et al., 2024c) further enhances RazorAttention by introducing a novel head assessment framework that simultaneously evaluates both retrieval and reasoning capabilities to optimize KV cache allocation strategies. DuoAttention (Xiao et al., 2024b) further introduces a parameterized approach to distinguish between two categories of attention mechanisms: retrieval heads, essential for comprehensive long-context processing, and Streaming heads, which primarily engage with recent tokens and attention sinks. This classification is achieved through learned parameters that automatically identify retrieval heads requiring full attention spans.

### 4.2.3   Summary and Future Directions

Despite recent advances and growing attention in KV cache budget allocation research, several critical challenges remain unaddressed. First, the relationship between allocation strategies and model performance requires further investigation. For instance, a notable discrepancy exists between pyramid-shaped allocation strategies (Cai et al., 2024; Yang et al., 2024a) advocating larger budgets for lower layers, and retrieval head-based studies (Tang et al., 2024a; Fu et al., 2024c) which demonstrate that lower layers rarely exhibit retrieval head characteristics and thus require minimal cache resources. Additionally, the field lacks

Table 4: The summary of existing KV Cache merging approaches.

| Model | Merge Layer | | Merge Unit | Merge Metric | Merge Type | Training-free |
| | Intra-layer | Cross-layer | | | | |
|---|:---:|:---:|:---:|:---:|:---:|:---:|
| **CCM** (Kim et al., 2024a) | ✓ | | Token | Sliding Window | Many-to-One | ✕ |
| **LoMA** (Wang & Xiao, 2024) | ✓ | | Token | Sliding Window | Many-to-Many | ✕ |
| **DMC** (Nawrot et al., 2024) | ✓ | | Token | Learned Merge Indictor | Many-to-One | ✕ |
| **D2O** (Wan et al., 2024a) | ✓ | | Token | Cosine Similarity | Two-to-One | ✓ |
| **CaM** (Zhang et al., 2024g) | ✓ | | Token | Attention Score | Many-to-One | ✓ |
| **AIM** (Zhong et al., 2024b) | ✓ | | Token | Cosine Similarity | Many-to-One | ✓ |
| **ZeroMerge** (Liu et al., 2025) | ✓ | | Token | Cosine Similarity | Many-to-One | ✓ |
| **Look-M** (Wan et al., 2024b) | ✓ | | Token | Cosine Similarity | Many-to-One | ✓ |
| **KVMerger** (Wang et al., 2024c) | ✓ | | Token | Weighted Gaussian Kernel | Many-to-One | ✓ |
| **CHAI** (Agarwal et al., 2024) | ✓ | | Head | Attention Score | Many-to-One | ✓ |
| **MinCache** (Liu et al., 2024a) | | ✓ | Token | Angular Distance | Two-to-One | ✓ |
| **KVSharer** (Yang et al., 2024d) | | ✓ | Layer | Euclidean Distance | Many-to-One | ✓ |

comprehensive experimental comparisons, particularly regarding the compatibility and performance benefits of head-wise budget allocation strategies with state-of-the-art frameworks like vLLM (Kwon et al., 2023) and FlashAttention (Dao et al., 2022; Dao, 2024; Shah et al., 2024). Also, existing methods, such as PyramidInfer (Yang et al., 2024a), demonstrate some adaptability to input attention patterns. However, future research could target real-time, task-specific allocation strategies that dynamically adjust memory budgets during inference based on input characteristics, task complexity, or downstream requirements.

## 4.3 KV Cache Merging

KV cache merging offers a promising solution by compressing or consolidating KV caches without significantly degrading model accuracy. Rather than a uniform compression strategy, KV cache merging techniques leverage the inherent redundancy within and across layers to dynamically optimize memory utilization. These methods aim to reduce the size of KV caches while preserving critical information necessary for accurate attention computations, enabling efficient inference in resource-constrained settings. Existing KV cache merging strategies can be categorized into two primary approaches: **intra-layer merging**, which focuses on consolidating KV caches within individual layers to reduce memory usage per layer, and **cross-layer merging**, which targets redundancy across layers to eliminate unnecessary duplication. Both approaches offer complementary advantages, providing flexibility to balance memory savings and model performance degradation. The summary of the KV cache merging is listed in Tab. 4.

### 4.3.1 Intra-layer Merging

As the input sequence length increases, the number of Keys and Values grows, leading to higher computational costs for the attention process. To address this, CCM (Kim et al., 2024a), LoMA (Wang & Xiao, 2024), DMC (Nawrot et al., 2024) propose to learn a compression module to compress KV of tokens.

Specifically, CCM (Kim et al., 2024a) inserts a special indicator token, `[COMP]`, into the input sequence and compresses the accumulating past attention key/value (KV) pairs in each layer between these indicators into a compact memory space. This compression leverages techniques inspired by the Compressive Transformer (Rae et al., 2020) and Gisting (Mu et al., 2023). Instead of computing attention across all tokens, CCM (Kim et al., 2024a) computes attention scores for each new token by referencing the merged token. Similarly, LoMA (Wang & Xiao, 2024) inserts a special token into the input sequence to determine

which consecutive tokens should be compressed. LoMA (Wang & Xiao, 2024) performs compression using bidirectional attention, repetition zone supervision, and carefully designed attention masks and loss functions. DMC (Nawrot et al., 2024) learns a variable to decide whether to append new KV pairs to the cache when necessary or to merge them into existing KV representations using a weighted average. Note that CCM (Kim et al., 2024a), LoMA (Wang & Xiao, 2024), and DMC (Nawrot et al., 2024) require supervised learning to learn a compression module.

Instead, CaM (Zhang et al., 2024g), KVMerger (Wang et al., 2024c), and D2O (Wan et al., 2024a) are training-free, which rely on observations and directly propose rule-based or heuristic-based merging strategies. Specifically, they separate the Keys and Values of tokens in each layer into important (retained) and unimportant (evicted) tokens. They then keep potentially useful unimportant tokens by merging their Keys and Values with retained important tokens, ensuring that no valuable information is lost. Particularly, D2O (Wan et al., 2024a) merges the Key (or Value) of a evicted token with one retained token based on cosine similarity. Similar to D2O, AIM (Zhong et al., 2024b), Look-M (Wan et al., 2024b), and ZeroMerge (Liu et al., 2025) merge Keys (resp. Values) of multiple tokens into one. CaM (Zhang et al., 2024g) merges the Keys (or Values) of multiple evicted tokens with retained tokens based on attention scores to get the final merged results. Also, KVMerger (Wang et al., 2024c) first identifies the merge token sets by clustering consecutive tokens with high cosine similarity, ensuring that only adjacent tokens with strong contextual relevance are grouped together. Then, KVMerger merges the tokens in each merge set into the pivotal token (chosen based on the highest attention score) using Gaussian kernel weights, where closer tokens contribute more to the merged state.

Instead of merging the KV of multiple tokens into one, CHAI (Agarwal et al., 2024) observes that heads in multi-head attention often produce highly correlated attention scores for tokens, particularly in the later layers of LLMs. To exploit this redundancy, CHAI (Agarwal et al., 2024) clusters attention heads within each layer that produce similar outputs and computes attention for only a single representative head in each cluster. Specifically, within each cluster, CHAI (Agarwal et al., 2024) selects one representative head to perform the attention computation, and the computed attention scores are shared across all heads in the cluster.

### 4.3.2 Cross-layer Merging

MiniCache (Liu et al., 2024a) observes that KV caches in middle-to-deep layers exhibit high angular similarity, making them ideal for merging. To achieve this, MiniCache (Liu et al., 2024a) merges the Key (and Value) pairs of each token from adjacent similar layers into a single shared representation. Specifically, Mini-Cache (Liu et al., 2024a) decomposes KV vectors into magnitude and direction components, storing only the shared directional vectors, token magnitudes, and unmergeable tokens to maximize memory efficiency. Differently, KVSharer (Yang et al., 2024d) observes a counterintuitive phenomenon: when the KV caches of two layers differ significantly, sharing one layer's KV cache with another during inference does not cause significant performance degradation. Based on this observation, KVSharer (Yang et al., 2024d) computes the Euclidean distance between the KV caches of all layer pairs, ranks the pairs by dissimilarity, and prioritizes the most dissimilar layers for sharing. Since KVSharer (Yang et al., 2024d) can share the KV cache of one layer to multiple other layers, the stored KV cache is eliminated significantly.

### 4.3.3 Summary and Future Directions

KV cache merging represents a transformative approach to optimizing memory utilization in LLMs by consolidating or compressing KV caches while maintaining high model accuracy. However, there are several key directions and challenges for future exploration in this domain. Firstly, current KV cache merging methods are typically designed to work across a wide range of tasks, but fine-tuning merging strategies for specific tasks or domains could further enhance efficiency. For example, certain tasks may tolerate more aggressive merging due to inherent redundancy in their attention patterns, while others may require more conservative approaches to preserve accuracy. Adaptive merging mechanisms that adjust compression levels on-the-fly based on task difficulty, sequence length, or available hardware resources are an exciting avenue for future work. Secondly, sparse attention mechanisms, which already reduce the computational complexity of attention by operating on subsets of tokens, could be combined with KV cache merging to achieve even greater efficiency. Exploring how merging complements sparsity-based approaches, such as

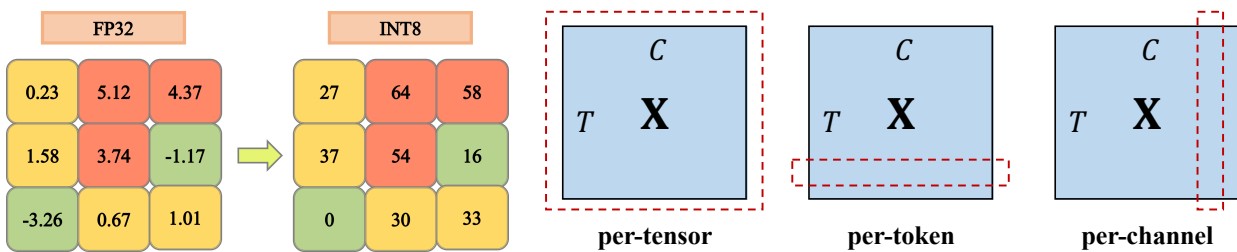

(a) The quantization of attention matrix.

(b) Three types of quantization. Then matrix $\mathbf{X} \in \mathbb{R}^{T \times C}$, where $T$ is the number of tokens and $C$ is the feature dimension.

Figure 4: An example of quantization and three types of quantization applied to each matrix.

block-sparse or low-rank attention, could lead to novel hybrid solutions. Thirdly, while empirical results show that merging does not significantly degrade performance, providing theoretical guarantees about the preservation of critical information could enhance the reliability of these methods. Future work might focus on quantifying the relationship between merging strategies, token importance, and attention accuracy to provide more formal guarantees.

### 4.4 KV Cache Quantization

As shown in Fig.4a, quantization techniques (Lin et al., 2016; Wu et al., 2020; Kwasniewska et al., 2019; Zhou et al., 2018; Jiang & Agrawal, 2018) aim to convert full-precision values into lower-bit integers, reducing computational and storage requirements. Quantization techniques have been widely used to accelerate machine learning models from different aspects, such model parameter quantization (Frantar et al., 2022; Dettmers et al., 2024; Bondarenko et al., 2023; Cheng et al., 2017) and data feature quantization (Zhou et al., 2023a; Jegou et al., 2010). Similarly, Key-Value (KV) cache quantization is emerging as a highly promising solution to address the memory and computational bottlenecks in LLMs. During auto-regressive decoding, LLMs generate key-value pairs for every attention layer across all tokens in the sequence. If we store all KV pairs in the memory with full precision, this cache grows linearly with longer sequences, increasing the memory and bandwidth requirements significantly. Quantization reduces the precision of numerical representations (e.g., from FP32 to INT8 or INT4), drastically compressing the size of the KV cache. This compression can achieve up to 4x or more memory savings, making it feasible for LLMs to operate on resource-constrained devices like GPUs with limited memory or edge devices.

However, the presence of outliers in Keys and Values poses a significant challenge for low-bit quantization, as these extreme values can lead to substantial performance degradation when compressed into reduced bit representations (Dettmers et al., 2022; Bondarenko et al., 2021; Wei et al., 2022). Based on the techniques used, existing KV cache quantization approaches can be classified into three main categories: **Fixed-precision quantization**, where all Keys and Values are quantized to the same bit-width; **Mixed-precision quantization**, which assigns higher precision to critical parts of the cache while using lower precision for less important components; and **Outlier redistribution**, which redistributes or smooths the outliers in Keys and Values to improve quantization quality. These methods collectively enable efficient KV cache compression while mitigating the performance degradation typically associated with low-bit quantization.

#### 4.4.1 Fixed-precision Quantization

Fixed-precision quantization proposes quantizing different Keys (different Values) of tokens to the same bit-width. ZeroQuant (Yao et al., 2022) proposes per-token quantization for Keys and Values. As shown in Fig. 4b, the per-token quantization approach quantize tokens individually. Particularly, ZeroQuant (Yao et al., 2022) dynamically computes the min-max range for each token during inference. This ensures that each token is quantized based on its unique range, significantly reducing quantization error. Also Flex-Gen (Sheng et al., 2023) and QJL (Zandieh et al., 2024) directly perform per-token quantization for Keys and Values, where the scaling factor and zero-point are shared among all elements within the same token. PQCache (Zhang et al., 2024b) uses product quantization approaches (Jegou et al., 2010; Matsui et al., 2018) to compress KV pairs. However, uniform quantization approaches, which use a fixed bit-width for

Table 5: The summary of existing mixed-precision quantization models.

| Model | Keys | Values | Important Tokens | | | Outlier storing | Channel Reorder |
|---|---|---|---|---|---|---|---|
| | | | Intial | Middle | Recent | | |
| **KVQuant** (Hooper et al., 2024b) | Channel, Pre-RoPE | Per-Token | ✓ | | | ✓ | |
| **KIVI** (Liu et al., 2024h) | Channel | Per-Token | | | ✓ | | |
| **SKVQ** (Duanmu et al., 2024) | Dynamic outlier-aware | | ✓ | | ✓ | | ✓ |
| **WKVQuant** (Yue et al., 2024) | Learnable shifting | | | | ✓ | | |
| **QAQ** (Dong et al., 2024b) | Adaptive quantization bits | | ✓ | ✓ | ✓ | ✓ | |
| **MiKV** (Yang et al., 2024c) | Dynamic outlier-aware | | ✓ | ✓ | ✓ | | |
| **GEAR** (Kang et al., 2024) | Dynamic outlier-aware | | | | ✓ | ✓ | |
| **ZIPVL** (He et al., 2024a) | Conventional | | ✓ | ✓ | ✓ | | |
| **CacheGen** (Liu et al., 2024f) | Layer-wise, token-locality | | | | | | |
| **Atom** (Zhao et al., 2024b) | Group-based | | | | | ✓ | ✓ |

keys and values across all tokens, can often be suboptimal. It is because they ignore the varying importance of tokens (Zhang et al., 2024f) and account for the outlier patterns in Keys and Values (Dong et al., 2024b; Hooper et al., 2024b).

### 4.4.2 Mixed-precision quantization

Unlike fixed-precision quantization, where all Keys or Values are quantized to the same bit-width (e.g., 4-bit or 8-bit), mixed-precision quantization assigns higher or full precision to Keys and Values of critical tokens and parts while using lower precision for less critical parts. The summary of KV mixed-precision quantization is listed in Tab. 5. KVQuant (Hooper et al., 2024b) proposes several strategies to quantize Keys and Values smoothly based on observations. Firstly, KVQuant observes that the key values exhibit outliers in specific channels prior to applying Rotary Positional Embedding (RoPE). However, after applying RoPE, the magnitudes of these outlier channels become less consistent, creating a unique challenge for low-precision quantization. Thus, KVQuant (Hooper et al., 2024b) proposes to quantize the Keys per channel before applying the RoPE operations and to quantize the Values per token. Secondly, KVQuant (Hooper et al., 2024b) observes that KV cache activations contain outliers that skew the quantization range. To address this, they isolate outliers per vector (e.g., per-channel or per-token), store them in a sparse format, and quantize the remaining values to a narrower range. Thirdly, LLMs disproportionately allocate high attention scores to the first token (i.e., attention sink), and quantizing the first token will damage the performance of LLMs. Thus, KVQuant (Hooper et al., 2024b) retains the first token in full precision (FP16) while quantizing the rest of the sequence, which is also used by IntactKV (Liu et al., 2024e) and SKVQ (Duanmu et al., 2024). Similar to KVQuant (Hooper et al., 2024b), KIVI (Liu et al., 2024h) quantizes the Key cache per-channel, as certain channels exhibit large outliers, and the Value cache per-token, as there are no significant outlier patterns in the Value cache. Additionally, KIVI (Liu et al., 2024h) retains the most recent Keys and Values in full precision, while quantizing older KVs. This approach is based on the observation that the most recent KVs are critical for generating subsequent tokens.

Similar to KIVI (Liu et al., 2024h), WKVQuant (Yue et al., 2024) temporarily retains the most recent Keys and Values in full precision, while quantizing only the past KV cache. This approach (Yue et al., 2024) helps preserve precision during computation. Similarly, ResQ (Saxena et al., 2024b) uses PCA to identify high-variance components, which are quantized in 8-bit, while the rest are quantized in 4-bit. Also, ResQ (Saxena et al., 2024b) uses random rotations to suppress outliers, improving robustness. Additionally, WKVQuant (Yue et al., 2024) introduces a two-dimensional quantization strategy, which optimizes the parameter matrix to align the values in the KV cache into a smoother and more uniform range, significantly improving quantization quality. GEAR (Kang et al., 2024), MiKV (Yang et al., 2024c), ZipCache (He et al.,

2024b) and ZIPVL (He et al., 2024a) quantize the KV cache based on the importance of each to achieve efficient and effective compression. First, GEAR (Kang et al., 2024) applies quantization to compress the majority of less important entries (e.g., 98%) to ultra-low precision, significantly reducing memory usage. Next, GEAR (Kang et al., 2024) employs a low-rank matrix to approximate residual errors, capturing structured patterns in the data. Also, GEAR (Kang et al., 2024) uses a sparse matrix to store outliers, correcting individual errors caused by these values. MiKV (Yang et al., 2024c) is a mixed-precision KV cache quantization approach. Based on the importance of each token, measured using existing methods like H2O (Zhang et al., 2023a) and SnapKV (Li et al., 2024b), MiKV (Yang et al., 2024c) stores less important KV pairs in low precision while retaining the most important KV pairs in high precision. Instead of approximating the importance weight of each token, ZipCache (He et al., 2024b) accurately computes the importance of each token. Instead of computing importance score, PrefixQuant (Chen et al., 2024c) observes that token-wise outliers frequently occur at fixed positions (e.g., initial tokens) or low-semantic-value tokens (e.g., ".", "\n"). Based on this observation, PrefixQuant (Chen et al., 2024c) identifies high-frequency outlier tokens in LLMs offline and prefixes them in the KV cache, effectively eliminating token-wise outliers. Similarly, MiniKV (Sharma et al., 2024) observes that important tokens can be identified before generation and remain consistent throughout the generation process, retaining these important tokens in high precision.

KV-AdaQuant (Hariri et al., 2025) observes the higher sensitivity of key matrices to quantization errors due to their larger norms, and therefore allocates more bits to key matrices and fewer bits to value matrices, optimizing both accuracy and memory efficiency. QAQ (Dong et al., 2024b) proposes a quality adaptive quantization approach to dynamically determine the suitable quantization bit for each token, based on its importance and sensitivity, while handling outliers and exceptions to maintain model performance. SKVQ (Duanmu et al., 2024) introduces the clipped dynamic quantization with channel reorder. First, SKVQ (Duanmu et al., 2024) uses a transformation-invariant permutation to group similar channels based on their statistical characteristics and applies clipped dynamic quantization to mitigate the outlier problem. Second, SKVQ (Duanmu et al., 2024) maintains high precision for the initial tokens and the most recent tokens while quantizing older tokens. Consequently, SKVQ (Duanmu et al., 2024) effectively reduces quantization errors and improves the accuracy of the quantized model. CacheGen (Liu et al., 2024f) and AsymKV (Tao et al., 2024) use layer-wise asymmetric quantization, assigning higher-bit precision to key matrices in sensitive early layers and lower-bit precision to less sensitive layers, balancing memory efficiency and performance. Particularly, CacheGen (Liu et al., 2024f) also exploits token-wise locality by encoding deltas (differences) between KV tensors of nearby tokens instead of raw values. Atom (Zhao et al., 2024b) identifies and separates outlier channels, reordering the matrix to group these outlier channels at the end, thereby ensuring regular memory access patterns for improved hardware utilization. Then, Atom (Zhao et al., 2024b) quantizes outliers with higher precision, while normal channels are quantized to INT4 for maximum efficiency. In particular, Atom (Zhao et al., 2024b) applies fine-grained group quantization by dividing matrices into smaller subgroups (e.g., 128 elements per group) and performing quantization independently within each group.

### 4.4.3 Outlier Redistribution

As previously mentioned, outliers in the Keys and Values present significant challenges for their quantization. Recent research has proposed two main approaches to address this issue: redistributing the outliers into newly appended virtual tokens or applying equivalent transformation functions to smooth the Keys and Values for improved quantization accuracy. The summary of existing outlier redistribution models are listed in Table. 6.

Specifically, MassiveActivation (Sun et al., 2024b) highlights the phenomenon of massive activations in large language models (LLMs), where a small subset of activations is exponentially larger than the rest. To address this, MassiveActivation (Sun et al., 2024b) proposes appending a virtual token to the inputs, allowing LLMs to encapsulate the massive outliers within these learned keys and values for each head. On the other hand, to further address this issue, existing researchers (Ashkboos et al., 2024; Lin et al., 2024d) introduce equivalent transformation function-based approaches. Equivalent transformation functions are mathematical transformations applied to activations that preserve the underlying information while redistributing or normalizing the values. By redistributing or scaling massive values, these approaches ensure that extreme outliers do not disproportionately affect downstream processes like compression or quantization. Firstly,

Table 6: The summary of outlier redistribution models in Sec. 4.4.3.

| Model | Operation | Formula | Learn | Remarks |
|---|---|---|---|---|
| **MassiveAct** (Sun et al., 2024b) | Add virtual tokens | $\text{softmax}\left(\dfrac{\mathbf{Q}\left[\mathbf{K}^T, \quad \mathbf{k}'\right]}{\sqrt{d}}\right)\begin{bmatrix}\mathbf{V}\\ \mathbf{v}'^T\end{bmatrix}$ | ✓ | Learnable $\mathbf{k}'$, $\mathbf{v}'$ |
| **QuaRot** (Ashkboos et al., 2024) | Hadamard rotation | $\mathbf{XW}^\top = (\mathbf{XH})(\mathbf{H}^\top\mathbf{W}^\top)$ | × | $\mathbf{H}^\top\mathbf{H} = \mathbf{I}$ |
| **Qserve** (Lin et al., 2024d) | Hadamard rotation | $\mathbf{XW}^\top = (\mathbf{XH})(\mathbf{H}^\top\mathbf{W}^\top)$ | × | $\mathbf{H}^\top\mathbf{H} = \mathbf{I}$ |
| **Q-INT4** (Xi et al., 2023) | Hadamard rotation | $\mathbf{XW}^\top = (\mathbf{XH})(\mathbf{H}^\top\mathbf{W}^\top)$ | × | $\mathbf{H}^\top\mathbf{H} = \mathbf{I}$ |
| **SmoothQuant** (Xiao et al., 2023) | Scaling | $(\mathbf{X}\operatorname{diag}(\mathbf{s})^{-1})\cdot(\operatorname{diag}(\mathbf{s})\mathbf{W}^\top)$ | × | $\mathbf{s} \in \mathbb{R}^{c_i}$ |
| **QS+** (Wei et al., 2023) | Scaling, Shifting | $((\mathbf{X}-\mathbf{z})\operatorname{diag}(\mathbf{s})^{-1}\cdot\operatorname{diag}(\mathbf{s})+\mathbf{z})\mathbf{W}^\top$ | × | $\mathbf{s} \in \mathbb{R}^{c_i}$ |
| **AWQ** (Lin et al., 2024c) | Scaling | $\arg\min_{\mathbf{s}}\left\|\mathbf{XW}^\top - \mathbf{X}\operatorname{diag}(\mathbf{s})^{-1})Q(\operatorname{diag}(\mathbf{s})\mathbf{W}^\top)\right\|$ | ✓ | Quantization $Q(\cdot)$ |
| **OmniQuant** (Shao et al., 2023) | Scaling, Shifting | $Q_a\left(\frac{\mathbf{X}-\delta}{\mathbf{s}}\right)Q_w\left(\mathbf{s}\odot\mathbf{W}^\top\right) + \mathbf{B} + \delta\mathbf{W}^\top$ | ✓ | Learnable $Q_a(\cdot)$, $Q_w(\cdot)$ |
| **DuQuant** (Lin et al., 2024b) | Rotation, Permutation | $[(\mathbf{X}\cdot\mathbf{\Lambda})\hat{\mathbf{R}}_{(1)}\cdot\mathbf{P}\cdot\hat{\mathbf{R}}_{(2)}]\cdot[\hat{\mathbf{R}}_{(2)}^\top\cdot\mathbf{P}^\top\cdot\hat{\mathbf{R}}_{(1)}^\top(\mathbf{\Lambda}^{-1}\cdot\mathbf{W}^\top)]$ | × | Matrices $\mathbf{P}$, $\mathbf{R}$ |
| **AffineQuant** (Ma et al., 2024b) | Affine transform | $\arg\min_{\mathbf{P}}\left\|\mathbf{XW}^\top - \mathbf{XP}^{-1}Q(\mathbf{PW}^\top)\right\|_F^2$ | ✓ | Quantization $Q(\cdot)$ |
| **FlatQuant** (Sun et al., 2024d) | Affine transform | AffineQuant + $\mathbf{P} = \mathbf{P}_1 \otimes \mathbf{P}_2$ | ✓ | Decomposition |

QuaRot (Ashkboos et al., 2024), Qserve (Lin et al., 2024d), and Q-INT4 (Xi et al., 2023) redistributes outlier values across all channels by Hadamard rotation, successfully lowering the maximum value of outlier tokens. The Hadamard rotation of activations can be incorporated into the preceding linear layer, thereby redistributing the outliers of Keys and Values into the parameters. Despite this improvement, outlier tokens still exhibit magnitudes hundreds of times greater than normal tokens, causing notable performance issues when using shared quantization scales across tokens (Chen et al., 2024c). Expanding on this idea, Spin-Quant (Liu et al., 2024g) proposes training an orthogonal matrix instead of relying on a random Hadamard matrix to achieve better performance. Similarly, DuQuant (Lin et al., 2024b) employs channel permutation to evenly distribute outliers across blocks and utilizes block rotation to further smooth outliers.

SmoothQuant (Xiao et al., 2023) leverages a key observation that different tokens show similar patterns of variation across their channels. Based on this insight, it strategically shifts the quantization complexity from activations to weights through an offline process. Specifically, SmoothQuant (Xiao et al., 2023) introduces a mathematically equivalent per-channel scaling transformation: $\mathbf{Y} = (\mathbf{X}\operatorname{diag}(\mathbf{s})^{-1})\cdot(\operatorname{diag}(\mathbf{s})\mathbf{W}) = \hat{\mathbf{X}}\hat{\mathbf{W}}$ where $\mathbf{X}$ represents Keys or Values, and the smoothing factor $\mathbf{s} \in \mathbb{R}^{C_i}$ is used to scale $\mathbf{X}$. This transformation achieves two key benefits: it smooths the distribution of Keys and Values to facilitate easier quantization, and it allows the smoothing factors to be efficiently incorporated into the parameters of previous layers during offline processing. In particular, the smooth factor $\mathbf{s}$ is dynamically decided on based on inputs. Similarly, The OS+ (Wei et al., 2023) introduces channel-wise shifting to eliminate outlier asymmetry and channel-wise scaling to reduce outlier concentration. These operations are seamlessly migrated to subsequent layers, maintaining equivalence with the floating-point model while improving quantization performance.

Instead of using handcrafted transformations (Lin et al., 2024b; Wei et al., 2023; Xiao et al., 2023) to shift the quantization difficulty from activations to weights, AffineQuant (Ma et al., 2024b) uses an affine transformation matrix that combines both scaling and rotation transformations. This allows it to optimize weight distributions more effectively, aligning them better with the quantization function and reducing quantization errors. The affine transformation matrix provides richer flexibility compared to SmoothQuant's scalar-based scaling, enabling finer adjustments to the weight and activation distributions. Based on AffineQuant (Ma et al., 2024b), FlatQuant (Sun et al., 2024d) introduces a fast and learnable affine transformation to enhance the flatness of weights and activations, which decomposes transformation into smaller matrices to reduce memory and computational costs. Similarly, AWQ (Lin et al., 2024c) and OmniQuant (Shao et al., 2023) propose differentiable and learnable equivalent transformations, which optimize the equivalent parameters (e.g., channel-wise scaling and shifting) in an end-to-end manner using gradient descent.

### 4.4.4 Summary and Future Directions

KV cache quantization is a crucial technique for reducing memory and computational overhead in large language models (LLMs) during auto-regressive decoding. While significant progress has been made, this field remains dynamic and rapidly evolving, with several promising directions for future research. Firstly, one promising avenue is the development of real-time adaptive quantization methods. These techniques could dynamically adjust quantization levels during inference based on real-time metrics such as token importance, outlier presence, or sequence length. Such an approach could significantly enhance efficiency while maintaining performance, especially for processing long sequences with varying levels of complexity. Secondly, another important direction is extending KV cache quantization to multi-modal and multi-task models. Multi-modal models, which process inputs from diverse domains such as text, vision, and audio, and multi-task scenarios often exhibit highly diverse attention patterns and memory demands. This necessitates the design of more advanced and tailored quantization strategies to balance efficiency and accuracy in these increasingly complex settings.

Thirdly, hybrid quantization techniques also hold significant potential. By combining fixed-precision, mixed-precision, and outlier redistribution methods, researchers could develop more versatile and efficient quantization frameworks. For instance, integrating mixed-precision allocation schemes with outlier smoothing transformations could optimize both memory usage and performance, offering a flexible approach adaptable to a variety of tasks and models. Finally, addressing the challenge of outliers remains a critical area of focus. Outliers can have a disproportionate impact on quantization efficiency and model performance. Future research could explore advanced outlier detection mechanisms or innovative encoding techniques to mitigate their effects. Improved handling of outliers could further enhance the effectiveness of quantization methods, enabling more robust and memory-efficient implementations.

## 4.5 KV Cache Low-rank Decomposition

Existing studies have demonstrated that the majority of information within KV caches can be captured by a small subset of their singular values or low-rank components, making low-rank decomposition a powerful tool for compression. By leveraging this property, KV cache low-rank decomposition techniques aim to reduce memory requirements while preserving the essential information required for accurate attention computations. Low-rank decomposition strategies can be classified into three main approaches: **Singular Value Decomposition (SVD)**, which exploits the low-rank structure of KV matrices to retain the most critical singular values; **Tensor Decomposition**, which factorizes KV matrices into smaller components for minimal redundancy; and **Learned Low-rank Approximation**, which incorporates adaptive mechanisms to optimize compression based on learned representations. Each method provides a unique balance of computational efficiency and accuracy retention, enabling scalable and memory-efficient LLM inference.

### 4.5.1 Singular Value Decomposition

Recent research has demonstrated that KV caches in large language models exhibit strong low-rank properties, where a small number of top singular values retain most of the information. Building upon this discovery, numerous low-rank approximation techniques have been proposed for KV cache optimization. These methods can be broadly categorized into two approaches: those that directly decompose the KV cache and those that apply low-rank approximations to the KV weight matrices.

Among the methods that directly decompose KV caches, ECKVH (Yu et al., 2024), EigenAttention (Saxena et al., 2024a), Q-Filters (Godey et al.), and ZDC (Zhang & Shen, 2024) all leverage Singular Value Decomposition (SVD) techniques but with distinct implementations. ECKVH (Yu et al., 2024) compresses KV caches by grouping attention heads, applying SVD, and retaining top singular values, which effectively reduces the number of KV heads while minimizing error. Similarly, EigenAttention (Saxena et al., 2024a) employs SVD to approximate keys, queries, and values with low-rank basis vectors, thereby reducing the dimensionality of KV matrices. Q-Filters (Godey et al.), meanwhile, offers a training-free KV cache compression method that utilizes SVD-based key projections for efficient and accurate attention score approximations.

Besides, Loki (Singhania et al., 2024) introduces an alternative approach that first computes approximate attention scores in a reduced dimensional space to efficiently rank and select top-k keys. It then calculates exact attention scores using only the selected keys in the transformed space, thereby reducing computational and memory requirements while maintaining performance. Furthermore, ZDC (Zhang & Shen, 2024) implements an adaptive hybrid compression ratio mechanism that assigns higher compression to less important tokens in shallower layers while preserving more important tokens in deeper layers, effectively leveraging the similarity of token characteristics in adjacent layers.

Taking a fundamentally different approach, LoRC (Zhang et al., 2024d) employs low-rank approximations of KV weight matrices rather than decomposing the KV pairs themselves. Specifically, LoRC uses SVD to compress the Keys and Values parameter matrices (i.e., $\mathbf{W}_i^k$ and $\mathbf{W}_i^v$), decomposing them as $\mathbf{W}_i^k = \mathbf{U}_i^k \mathbf{\Sigma}_i^k \mathbf{P}_i^{k\top}$ and $\mathbf{W}_i^v = \mathbf{U}_i^v \mathbf{\Sigma}_i^v \mathbf{P}_i^{v\top}$. Additionally, LoRC adopts a progressive compression strategy, applying compression conservatively in shallower layers to minimize error amplification while compressing more aggressively in deeper layers. Instead of storing complete $\mathbf{K}^i = \mathbf{X}_i \mathbf{W}_i^k$ and $\mathbf{V}^i = \mathbf{X}_i \mathbf{W}_i^v$ matrices, it only stores $\hat{\mathbf{K}}^i = \mathbf{X}_i \mathbf{U}_i^k$ and $\hat{\mathbf{V}}^i = \mathbf{X}_i \mathbf{U}_i^v$, along with $\mathbf{\Sigma}_i^k \mathbf{P}_i^{k\top}$ and $\mathbf{\Sigma}_i^v \mathbf{P}_i^{v\top}$, achieving efficient KV cache compression without requiring model retraining. Palu (Chang et al., 2024) follows a comparable approach by applying SVD to compress KV weight matrices simultaneously, while ShadowKV (Sun et al., 2024a) takes a different angle by performing SVD decomposition directly on pre-RoPE keys to reduce the dimensionality of key representations.

### 4.5.2 Tensor Decomposition

Tensor decomposition (Kuleshov et al., 2015; Zhou et al., 2017; Haeffele & Vidal, 2015) is a widely used algorithm for factorizing a matrix into a sequential product of local tensors, such as Matrix Product Operator (MPO) (Liu et al., 2021) and turker decomposition (Malik & Becker, 2018). Taking Matrix Product Operator (MPO) (Liu et al., 2021) as an example, the decomposition of a matrix $\mathbf{W} \in \mathbb{R}^{I \times J}$ using MPO can be defined as:

$$\text{TD}(\mathbf{W}) = \prod_{k=1}^{n} \mathcal{T}_{(k)}[d_{k-1}, i_k, j_k, d_k], \tag{12}$$

where $\mathcal{T}_{(k)}$ represents the local tensor of size $d_{k-1} \times i_k \times j_k \times d_k$, with $\prod_{k=1}^{n} i_k = I$ and $\prod_{k=1}^{n} j_k = J$. Here, $n$ denotes the number of local tensors, collectively referred to as the decomposed tensors. As shown in Equation 12, MPO-based tensor decomposition is well-suited for KV cache compression as it reduces the memory footprint by factorizing large key and value matrices into smaller local tensors, enabling efficient storage while preserving essential information. This approach minimizes redundancy and maintains the structural integrity required for accurate attention computations. DecoQuant (Liu et al., 2024d) combines quantization with low-rank decomposition to effectively reduce quantization errors. Specifically, DecoQuant (Liu et al., 2024d) leverages the Matrix Product Operator (MPO) to decompose matrices into smaller local tensors. The larger tensors, which contain most of the parameters, are quantized to low-bit precision, while the smaller tensors retain high precision to minimize overall quantization error.

### 4.5.3 Learned Low-rank Approximation

LESS (Dong et al., 2024a) introduces a novel learned-kernel-based low-rank approximation approach to efficiently approximate the results of the softmax function. Specifically, LESS (Dong et al., 2024a) replaces the softmax with a separable similarity metric, $\phi(\mathbf{q}_t)\psi(\mathbf{K}_t)^\top$, where $\phi$ and $\psi$ are row-wise functions. Here, $\mathbf{q}_t \in \mathbb{R}^{1 \times D}$ represents the query, and $\mathbf{K}_t \in \mathbb{R}^{t \times D}$ represents the keys at step $t$. To elaborate, if $\phi$ and $\psi$ are such that: $a_t = \text{softmax}\left(\frac{\mathbf{q}_t \mathbf{K}_t^\top}{\sqrt{D}}\right)\mathbf{V}_t \approx \frac{\phi(\mathbf{q}_t)\psi(\mathbf{K}_t)^\top \mathbf{V}_t}{\phi(\mathbf{q}_t)\psi(\mathbf{K}_t)^\top \mathbf{1}_{S \times 1}}$, then we only need to cache the hidden states $\mathbf{H}_t = \psi(\mathbf{K}_t)^\top \mathbf{V}_t \in \mathbb{R}^{R \times D}$ and the normalization factor $\mathbf{z}_t = \sum_{s=1}^{t} \psi([\mathbf{K}_t]_s) \in \mathbb{R}^{1 \times R}$ for inference. Similarly, MatryoshkaKV (Lin et al., 2024a) compresses KV caches along the feature dimension by leveraging trainable orthogonal projection matrices.

### 4.5.4 Summary and Future Directions

KV cache low-rank decomposition is a powerful technique for compressing KV caches in LLMs while maintaining the quality of attention computations. Current methods primarily rely on fixed low-rank approximations

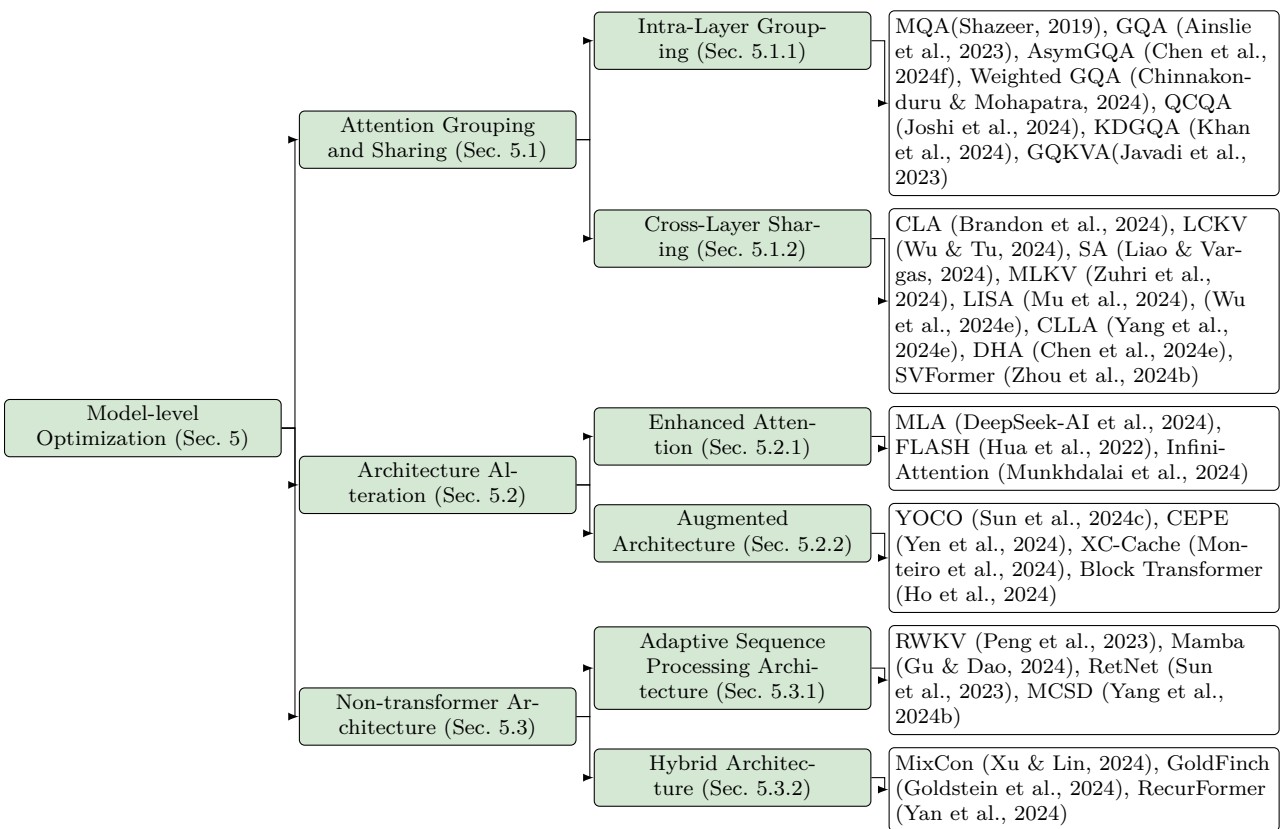

Figure 5: Taxonomy of the model based KV optimization for Large Language Models.

applied uniformly across all layers or tokens. However, future advancements could focus on dynamic rank adjustment, where the rank is tailored based on token importance, sequence length, or layer-specific properties, enabling a more optimal balance between memory efficiency and performance. Additionally, real-time or streaming applications present a promising avenue for exploration. Since KV caches grow dynamically during inference, lightweight and incremental decomposition methods that can adapt efficiently to expanding sequences will be critical for supporting such scenarios without compromising latency or accuracy.

## 5    Model-level Optimization

In model-level optimization, new architectures or mechanisms are designed for transformers to allow more efficient reuse of KV cache. Typically, these methods require retraining or fine-tuning of the model to come into operation. Nevertheless, efficient transformation pipelines have also been proposed to allow for a fast deployment to new architectures. We categorize related works according to their grouping and sharing mechanisms, either within layers or across layers (Sec. 5.1), implementing architecture modification or augmentation (Sec. 5.2), and incorporating non-transformer architectures for optimization (Sec. 5.3). The taxonomy of the model-level optimization is shown in Fig. 5.

### 5.1    Attention Grouping and Sharing

This section explores attention grouping and sharing methods as effective strategies for optimizing key-value (KV) management. We categorize the approaches into two distinct subtypes: intra-layer grouping (Sec. 5.1.1) that focuses on grouping query, key, and value heads within individual layers to reduce redundancy and improve efficiency, and cross-layer sharing (Sec. 5.1.2) that shares key, value, or attention components across

Table 7: The summary of Model-based Attention Grouping and Sharing approaches.

| Method | Applied Location | | Intra-layer Grouped Component | Cross-layer Shared Component | Retraining Required |
|---|---|---|---|---|---|
| | Intra-layer | Cross-layer | | | |
| **MQA** (Shazeer, 2019) | ✓ | | K, V | - | ✓ |
| **GQA** (Ainslie et al., 2023) | ✓ | | K, V | - | Uptrain |
| **AsymGQA** (Chen et al., 2024f) | ✓ | | K, V | - | Finetune |
| **Weighted GQA** (Chinnakonduru & Mohapatra, 2024) | ✓ | | K, V | - | Uptrain & Finetune |
| **QCQA** (Joshi et al., 2024) | ✓ | | K, V | - | ✓ |
| **KDGQA** (Khan et al., 2024) | ✓ | | K, V | - | ✓ |
| **GQKVA** (Javadi et al., 2023) | ✓ | | Q, K, V | - | ✓ |
| **CLA** (Brandon et al., 2024) | ✓ | ✓ | K, V | K, V | ✓ |
| **LCKV** (Wu & Tu, 2024) | | ✓ | - | K, V | ✓ |
| **SA** (Liao & Vargas, 2024) | | ✓ | - | Attention Weight | ✓ |
| **MLKV** (Zuhri et al., 2024) | ✓ | ✓ | K, V | K, V | Uptrain |
| **LISA** (Mu et al., 2024) | | ✓ | | Q, K, V | Lightweight adaption |
| **Wu et al.** (Wu et al., 2024e) | | ✓ | - | Q, K, V | ✓ |
| **CLLA** (Yang et al., 2024e) | | ✓ | - | Q, K, V | ✓ |
| **DHA** (Chen et al., 2024e) | ✓ | ✓ | K, V | Q, K, V | Lightweight adaption |
| **SVFormer** (Zhou et al., 2024b) | | ✓ | - | V | ✓ |

layers to improve information reuse and reduce KV cache requirements. The summary of attention grouping and sharing is listed in Tab. 7.

### 5.1.1 Intra-layer Grouping

As shown in Fig. 6a, Shazeer first introduced Multi-Query Attention (MQA) (Shazeer, 2019) that modified the traditional multi-head attention mechanism. In MQA, all attention heads in a transformer block share a single key and value. This simple strategy can greatly accelerate the decoding procedure. The experiments of the author show that MQA would gain much efficiency with only minor quality degradation occurring.

MQA is a radical strategy that would cause not just quality degradation, but also training instability. GQA (Grouped Query Attention) (Ainslie et al., 2023) introduced a trade-off solution by dividing the query heads into multiple groups, while each group shares its own keys and values. In addition, an uptraining process is proposed to efficiently convert existing MHA models to GQA configurations by mean-pooling the key and value heads associated with each group. Empirical evaluations demonstrated that GQA models achieve performance close to the original MHA models while offering inference time comparable to MQA.

There are several extensions based on GQA. AsymGQA (Chen et al., 2024f) extends GQA by proposing an activation-informed merging strategy. Instead of grouping the heads by uniform clustering, AsymGQA dynamically determines the grouping of queries based on their activation similarities during training and constructs symmetric group results, which leads to better optimization and generalization. Weighted GQA (Chinnakonduru & Mohapatra, 2024) introduces additional trainable weights to each key and value head,

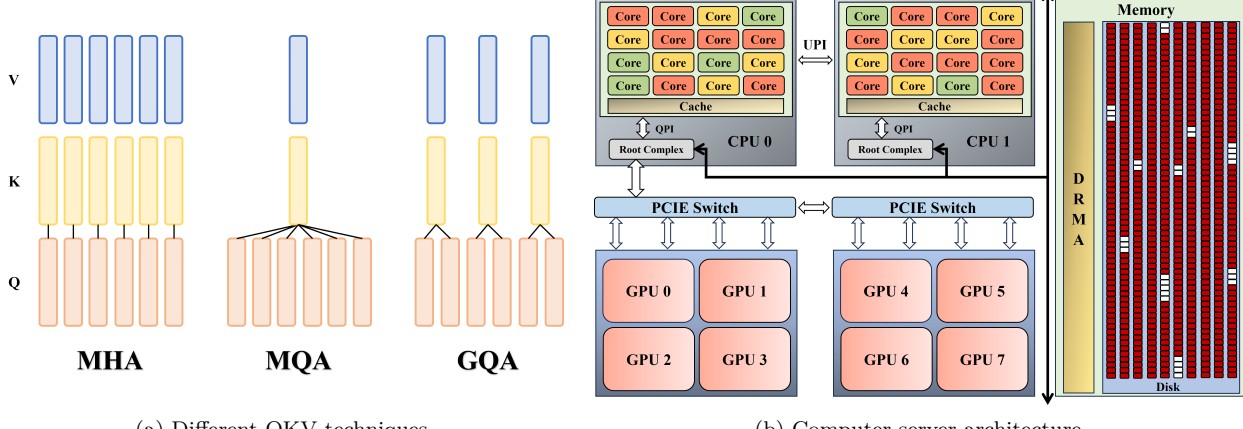

(a) Different QKV techniques.

(b) Computer server architecture.

Figure 6: Three types of QKV and a system architecture.

which can be seamlessly integrated into existing GQA models. By tuning weights during training, it improves the performance of the model without additional inference overhead. QCQA (Joshi et al., 2024) utilizes an evolutionary algorithm to identify the optimal query head groupings for GQA, which is guided by a computationally efficient fitness function that leverages the weight-sharing error and the KV cache to evaluate text generation quality and memory capacity. KDGQA (Khan et al., 2024) argues that many variants of GQA adopt a fixed grouping strategy, thus lacking dynamic adaptability to the evolving of key-value interactions during training. Their Dynamic Key-Driven GQA addresses these issues by adaptively allocating groups using key head norms during training, resulting in a flexible strategy for query head grouping that enhances performance.

GQKVA (Javadi et al., 2023) advances the grouping strategy and comes up with a generalized query, key and value grouping mechanism. It first introduces MKVA and GKVA, in which the key and value are grouped to share the same query. Based on this, GQKVA is proposed to separately group the query and key-value pairs. Typically, queries are partitioned into $g_q$ groups, and keys and values are partitioned into $g_{kv}$ groups, and each combination of query and key-value pairs would interact using dot product attention. This results in $g_q \times g_{kv}$ distinct outputs. It generalizes different group strategies on query, key and value and preserves good computational efficiency and comparable performance as MHA.

### 5.1.2 Cross-layer Sharing

Brandon et al. introduce Cross Layer Attention (CLA) (Brandon et al., 2024) that extends the ideas of GQA and MQA by sharing the key and value heads between adjacent layers, further reducing the redundancy in the KV cache. This achieves an additional 2× KV cache size reduction compared to MQA, significantly improving memory efficiency without altering computational complexity.

LCKV (Wu & Tu, 2024) proposes only to compute and cache the key and value for a small subset of layers, even only the top layer, then let queries in bottom layers pair the saved keys and values for inference. This method not only drastically improves the inference speed and reduces memory consumption but is also orthogonal to existing memory-saving techniques, enabling straightforward integration for further optimization. While such a mechanism makes next token computation depend on top layer keys and values of previous tokens, which contradicts the parallel training of transformers, LCKV introduces an approximate training method to support parallel training.

SA (Shared Attention) (Liao & Vargas, 2024) proposes reuse of computed attention weights across multiple layers, rather than recalculating them for each layer. Unlike other methods focusing on sharing key-value caches, SA leverages the isotropic tendencies of attention distributions observed in pre-trained LLMs to directly share attention weights, greatly reducing both computational overhead and memory usage.

MLKV (Multi-Layer Key-Value) (Zuhri et al., 2024) introduces a simple KV head sharing mechanism across multiple transformer layers. MLKV uses the same single KV head as MQA within a layer, but it also shares this KV head with multiple layers. This extreme strategy reduces the cache size to almost 1% of normal GQA strategies, and experiments show that MLKV still has comparable performance.

LISA (Lightweight Substitute for Attention) (Mu et al., 2024) makes a comprehensive analysis of the similarity of attention patterns across layers. Directly sharing attention weights across layers is ineffective because of the misalignment of the attention head and the sensitivity of shallow layers. LISA (Mu et al., 2024) addresses challenges by incorporating tiny feed-forward networks to align attention heads between layers and using low-rank matrices to approximate variations in layer-wise attention weights. This achieves a $6\times$ compression of query and key parameters while maintaining high accuracy and perplexity.

Wu et al. (2024e) introduce a unified framework that systematically analyzes and optimizes the cross-layer Key-Value cache sharing mechanism. They consolidate several existing methods, explore novel variants within a cohesive structure, and make thorough evaluations of these methods. The study finds that two times reduction to KV cache size can outperform standard transformers in throughput without substantial accuracy loss, while further reduction requires alternative design with additional training costs. With the analysis results, they offer insight into the choice of appropriate KV sharing methods based on the specific requirements or constraints.

CLLA (Cross-Layer Latent Attention) (Yang et al., 2024e) introduces an integrated framework combining multiple strategies: attention head size and dimension reduction, cross-layer cache sharing, and KV cache quantization. By unifying these strategies, CLLA achieves extreme KV cache compression to less than 2% of the original model size while maintaining performance levels comparable with uncompressed models.

DHA (Decoupled Head Attention) (Chen et al., 2024e) addresses redundancy in MHA and adaptively configures shared groups for key and value heads across layers, reducing KV cache requirements. Observing that clustering and fusing similar heads can reduce KV cache size without significant performance reduction, DHA designs a search, fusion, and continued pre-training framework that can progressively transform MHA checkpoints into DHA models through linear fusion of head parameters, preserving the pre-trained knowledge with a small pre-training budget.

Observing that later layers in traditional transformers overly rely on narrow regions of attention, Zhou et al. (2024b) introduce ResFormer that utilizes residual connections from the value embeddings of the first layer to all subsequent layers, effectively approximating cross-layer attention without incurring significant computational costs. They then propose a simplified variant SVFormer that shares a single value embedding across all layers, dramatically reducing the KV cache size by nearly half while maintaining competitive performance. The proposed architectures are flexible to incorporate with other KV-efficient strategies for additional memory savings.

### 5.1.3 Summary and Future Directions

This section explores innovative strategies for optimizing memory and computational efficiency through intra-layer grouping and cross-layer sharing mechanisms, while identifying key challenges and future directions. Maintaining performance for precision-sensitive tasks, ensuring scalability across diverse model architectures, and addressing the under-explored dynamics of attention across time and layers remain critical areas for improvement. Current approaches, like DHA (Chen et al., 2024e) and LISA (Mu et al., 2024), often struggle to generalize to emerging or non-standard architectures, while static grouping and sharing strategies fail to capture temporal and contextual attention variations. Future research should focus on developing universal, adaptable frameworks that require minimal retraining, integrating optimization techniques like quantization and pruning, and leveraging dynamic, runtime adjustments to better capture task-specific requirements. Additionally, understanding the downstream impacts on fine-tuning and transfer learning is essential for real-world applicability.

Table 8: The summary of Model-based Intra-layer approaches.

| Method | Alteration Type | | KV Cache Management | Retraining Requirement |
|---|---|---|---|---|
| | Enhanced Attention | Augmented Architecture | | |
| **MLA** (DeepSeek-AI et al., 2024) | ✓ | | Latent compression | ✓ |
| **FLASH** (Hua et al., 2022) | ✓ | | Linear approximation | ✓ |
| **Infini-Attention** (Munkhdalai et al., 2024) | ✓ | | Compressive cache | ✓ |
| **YOCO** (Sun et al., 2024c) | | ✓ | Single global KV cache | ✓ |
| **CEPE** (Yen et al., 2024) | | ✓ | Parallel encoding with cross-attn | Lightweight |
| **XC-Cache** (Monteiro et al., 2024) | | ✓ | Encoder cross-attention | ✓ |
| **Block Transformer** (Ho et al., 2024) | | ✓ | Hierarchical local KV | Lightweight |

## 5.2 Architecture Alteration

This section explores architectural modifications to optimize KV cache usage. We categorize these methods into two subsections: methods that refine the attention mechanism for KV cache efficiency (Sec. 5.2.1), and methods that introduce structural changes for better KV management (Sec. 5.2.2). Many of these works build upon the broader landscape of efficient attention mechanisms (e.g., Linear Transformer (Katharopoulos et al., 2020), Performer (Choromanski et al., 2020), LinFormer (Wang et al., 2020), etc.). Since our focus lies on methods directly impacting KV cache handling, for a comprehensive overview of efficient attention mechanisms, we refer readers to dedicated surveys (Zhou et al., 2024c). The summary of architecture alteration for KV reuse is listed in Tab. 8.

### 5.2.1 Enhanced Attention

DeepSeek-V2 (DeepSeek-AI et al., 2024) introduced Multi-Head Latent Attention (MLA) that adopts a low-rank KV joint compression mechanism, replacing the full KV cache with compressed latent vectors. The model adopts trainable projection and expansion matrices to do the compression. This compression mechanism significantly reduces the memory requirements of the KV cache and enables the model to handle sequences of up to 128K tokens.

FLASH (Hua et al., 2022) incorporates the Gated Attention Unit (GAU) to replace the MHA mechanism in traditional transformers. GAU employs a single-head attention mechanism with gating functions that selectively modulate the importance of information flow. FLASH employs a linear approximation method for attention computation through GAU module, which makes the model efficiently handle long contexts without the quadratic scaling of traditional self-attention, thus mitigating heavy KV cache issues.

Infini-Attention (Munkhdalai et al., 2024) adopts representation compression to store long-term content. Furthermore, they introduce a hybrid attention mechanism of masked local attention and long-term linear attention. The masked local attention replaces the standard MHA to let the model only concentrate on local contexts, while the long-term linear attention utilizes compressed memory for far-reaching dependencies and uses linear attention for efficient aggregation. Thus, infini-attention combines both local fine-grained and long-range compressed states, allowing a seamless balance between long-term and short-term context modeling.

Table 9: The summary of Non-Transformer Architectures.

| Method | Key Mechanism | No Traditional KV Cache | KV Compression |
|---|---|:---:|:---:|
| **RWKV** (Peng et al., 2023) | RNN-like with Transformer parallelism | ✓ | |
| **Mamba** (Gu & Dao, 2024) | Selective state-space model | ✓ | |
| **RetNet** (Sun et al., 2023) | Retention mechanism | | ✓ |
| **MCSD** (Yang et al., 2024b) | Slope-decay fusion | ✓ | |
| **MixCon** (Xu & Lin, 2024) | Transformer + Conba + MoE | ✓ | |
| **GoldFinch** (Goldstein et al., 2024) | RWKV + Modified Transformer | | ✓ |
| **RecurFormer** (Yan et al., 2024) | Mamba replacing some attention heads | | ✓ |

### 5.2.2 Augmented Architecture

YOCO (Sun et al., 2024c) builds a decoder-decoder architecture composed of two modules: a self-decoder and a cross-decoder. The self-decoder efficiently encodes global key-value caches, while the cross-decoder reuses these caches via cross-attention. This design ensures that key-value pairs are only cached once, substantially reducing GPU memory usage while maintaining global attention capabilities. YOCO's computation flow also enables the prefilling to early exit, allowing faster prefill stages without altering the final output.

CEPE (Yen et al., 2024) interleaves additional cross-attention layers between the self-attention and feed-forward layers in the decoder model. It employs a small encoder to process long inputs chunk-by-chunk to encoded representations as cross-attention layers' inputs. In this way, CEPE can prevent the need for KV cache for every token and reduce computational cost by processing contexts in parallel. This also facilitates an existing LLM to expand its contexts while preserving the scalability and generalizability.

XC-Cache (Monteiro et al., 2024) also utilizes an encoder to interleave cross-attention layers within existing self-attention layers in pre-trained decoder-only models to prevent explicit prompt caching. The encoder processes the context and converts it into a compact set of key-value pairs that summarize the essential information. It also finds that pre-trained causal decoders can be used to replace an encoder for the representations extraction, further reducing the training costs on an additional encoder.

Block Transformer (Ho et al., 2024) introduces a hierarchical global-to-local architecture by combining coarse-grained global attention and fine-grained local attention. In lower layers, tokens are grouped into fixed-size blocks, allowing global context modeling with reduced KV cache overhead. In upper layers, attention operates within individual blocks, enabling lightweight, detailed token decoding with a smaller local KV cache.

### 5.2.3 Summary and Future Directions

This section explores research that introduces novel attention mechanisms or architectural modifications to improve KV cache management. Although these approaches demonstrate significant progress in enabling longer context windows and faster inference, several challenges remain. First, many methods, such as CEPE (Yen et al., 2024) and XC-Cache (Monteiro et al., 2024) demonstrate strong performance on retrieval-augmented tasks but may not generalize well across diverse workloads. This necessitates further research into task-adaptive KV cache optimization strategies that dynamically adjust caching behavior to optimize for different task demands. Secondly, integrating these novel mechanisms into existing pretrained models often requires extensive retraining, hindering their adoption in resource-constrained environments. Developing lightweight, modular approaches for retrofitting efficient KV caching into existing architectures is crucial for a wider practical impact. Finally, the robustness and stability of these new mechanisms under real-world conditions, such as noisy or dynamically changing inputs, require further investigation. Addressing these limitations could improve reliability and efficiency in practical deployments.

### 5.3 Non-Transformer Architecture

While transformers are struggling with KV cache issues, researchers have revisited principles from traditional sequential architectures, such as recurrent neural networks (RNNs) (Salehinejad et al., 2017), which inher-

ently process sequences without the need for explicit KV caches. Inspired by the lightweight and memory-efficient design of RNNs and efficient attention mechanisms, non-transformer architectures (Xu et al., 2024e; Hasani et al., 2022; Smith et al., 2022; Wang et al., 2022; Gu & Dao, 2024; Peng et al., 2023) have emerged, such as Mamba (Gu & Dao, 2024) and RWKV (Peng et al., 2023), offering promising alternatives. While there are a large type of new architectures, we only list methods associated with KV optimization. For further understanding of efficient non-transformer works, please refer to these surveys (Zhou et al., 2024c; Xu et al., 2024d; Qu et al., 2024; Patro & Agneeswaran, 2024). The summary of non-transformer is listed in Tab. 9.

### 5.3.1 Adaptive Sequence Processing Architectures

RWKV (Peng et al., 2023), which means Receptance Weighted Key Value, is an architecture that combines the strengths of RNNs and transformers to achieve efficient sequence processing. RWKV integrates a linear attention mechanism, enabling parallelizable training like transformers while retaining the efficient inference characteristics of RNNs. By formulating the architecture to operate as either a transformer or an RNN, RWKV achieves constant computational and memory complexity during inference, overcoming the quadratic scaling issues of transformers.

Mamba (Gu & Dao, 2024) is built based on state space sequence models (SSMs) (Gu et al., 2022; 2021). Inspired by the state space systems, SSMs build scalable and memory-efficient long-range sequence modeling frameworks. Mamba improves SSMs by making parameters input-dependent, allowing information to be selectively propagated or forgotten along the sequence based on the current token. This addresses the inability of traditional SSMs to effectively handle the complexity of nonlinear dependencies in natural languages. Mamba omits attention and even MLP blocks, relying entirely on these selective state spaces for sequence modeling. It also develops a hardware-aware parallel algorithm for efficient recurrent computations in training and inference. Mamba achieves linear scaling in sequence length, demonstrating exceptional performance on sequences of up to a million tokens.

RetNet (Sun et al., 2023) introduces the Retentive Network, which combines elements of recurrence and attention, presenting a novel retention mechanism for sequence modeling that enables training parallelism, low-cost inference, and scalable performance. The proposed Multi-scale Retention Module (MSR) supports multiple computation paradigms: the parallel representation, similar to self-attention, allows for causal masking and parallel training; the recurrent representation, akin to RNNs, enables low-cost inference by maintaining state across sequence decoding; and the chunkwise recurrent representation constructs a hybrid of the previous two approaches, further enhancing the ability to handle long sequences.

MCSD (Yang et al., 2024b) introduces a new block called Multi-Channel Slope and Decay, which consists of two sections: the slope section, which captures local features across short temporal spans, and the decay section, which captures global features across long temporal spans. These sections are fused through element-wise operations. During inference, the process is reformulated into a recurrent representation, allowing for both spatial and temporal efficiency and minimizing the need to maintain a large KV cache.

### 5.3.2 Hybrid Architecture

With these non-transformer architectures, some methods construct mixed models to alleviate KV cache necessities while keeping some peculiarities and merits of the self-attention mechanism.

MixCon (Xu & Lin, 2024) introduces a new architecture called Conba. Inspired by control theory, the Conba layer incorporates a feedback and adaptive control mechanism that can adapt to different sequence-modeling tasks and requirements dynamically with good computational efficiency. Furthermore, MixCon integrates the Mixture of Experts (MoE) module, which dynamically selects the most relevant experts to process parts of the sequence. Combining the transformer layer, the Conba layer, and the MoE module, MixCon constructs a hybrid model with a good balance between attention effectiveness and computational efficiency and significantly reduces the total size of the KV cache.

GoldFinch (Goldstein et al., 2024) first introduces several new architectures, including the GOLD layer, which combines the Llama and RWKV channel mixer with several improvements, and the enhanced Finch model (RWKV-6) that has significantly reduced parameters without sacrificing efficiency and performance.

GoldFinch also proposes a novel mechanism called TokenCat to produce a highly compressed global key cache using the output of Finch layers. GoldFinch builds a hybrid architecture that constructs the key cache in the early layers and consumes the key cache to produce output without the traditional value cache in the top layers, providing a compact and reusable cache pipeline with linear scaling.

RecurFormer (Yan et al., 2024) argues that not all transformer heads need to participate in the self-attention mechanism. The authors observe that certain attention heads exhibit recency-aware behavior, focusing on local and short-range dependencies. These heads consume computational resources but contribute little to overall performance. After identifying such heads, RecurFormer replaces them with Mamba components, resulting in straightforward KV cache reduction.

### 5.3.3 Summary and Future Directions

By exploring non-transformer modules such as recurrent and hybrid designs, these methods have introduced novel paradigms that balance performance with computational efficiency, and also alleviate the KV cache issues in traditional transformer architectures. Future research should focus on several key areas. First, improving the scalability of recurrent architectures, such as RWKV (Peng et al., 2023) and Mamba (Gu & Dao, 2024), remains critical. Although these methods reduce memory and computational costs, their performance in capturing ultra-long-range dependencies lags behind transformers. Second, hybrid designs such as MixCon (Xu & Lin, 2024) and GoldFinch (Goldstein et al., 2024) highlight the potential of integrating diverse modules, yet their complexity introduces challenges in training stability and interpretability. Third, the overall generalization capabilities and robustness of non-transformer architectures need exploration for diverse input modalities.

## 6 System-level Optimization

As shown in Fig. 6b, existing computing server systems consist of various components, such as GPUs, CPUs, and memory storage. Optimizing LLM acceleration via KV cache at the system level is both practical and intriguing, considering the different communication and data exchange mechanisms. Recent system-level optimizations for KV cache in LLM inference can be broadly categorized into three main directions: memory management (Sec. 6.1), scheduling strategies (Sec. 6.2), and hardware-aware designs (Sec. 6.3). These complementary approaches collectively demonstrate the rich design space for system-level optimizations in LLM inference, each addressing different aspects of the performance, efficiency, and resource utilization challenges. The taxonomy of the system-level optimization is in Fig. 7.

### 6.1 Memory Management

Recent advances in KV cache memory management for large language model (LLM) inference reveal three distinct approaches aimed at enhancing memory efficiency. Architectural designs, exemplified by vLLM with PagedAttention (Kwon et al., 2023) and vTensor (Xu et al., 2024c), adapt classical operating system principles to create flexible, dynamic memory allocation systems that optimize the use of physical memory through sophisticated mapping and virtual memory abstractions. Prefix-aware designs like ChunkAttention (Ye et al., 2024) and MemServe (Hu et al., 2024) further refine this approach by organizing data structures to enable efficient cache de-duplication and sharing of common prefixes, thereby improving both memory utilization and computational efficiency. Together, these innovations illustrate the potential for significant enhancements in LLM serving via memory management.

### 6.1.1 Architectural Design

The first category focuses on architectural innovations in memory management, led by vLLM with PagedAttention (Kwon et al., 2023), which adapts OS-inspired paging concepts by partitioning KV caches into fixed-size blocks with non-contiguous storage. PagedAttention partitions KV caches into fixed-size blocks that can be stored non-contiguously in physical memory, while vLLM (Kwon et al., 2023) implements a virtual memory-like system that manages these blocks through a sophisticated mapping mechanism. This architecture separates logical and physical KV blocks, enabling dynamic memory allocation and flexible

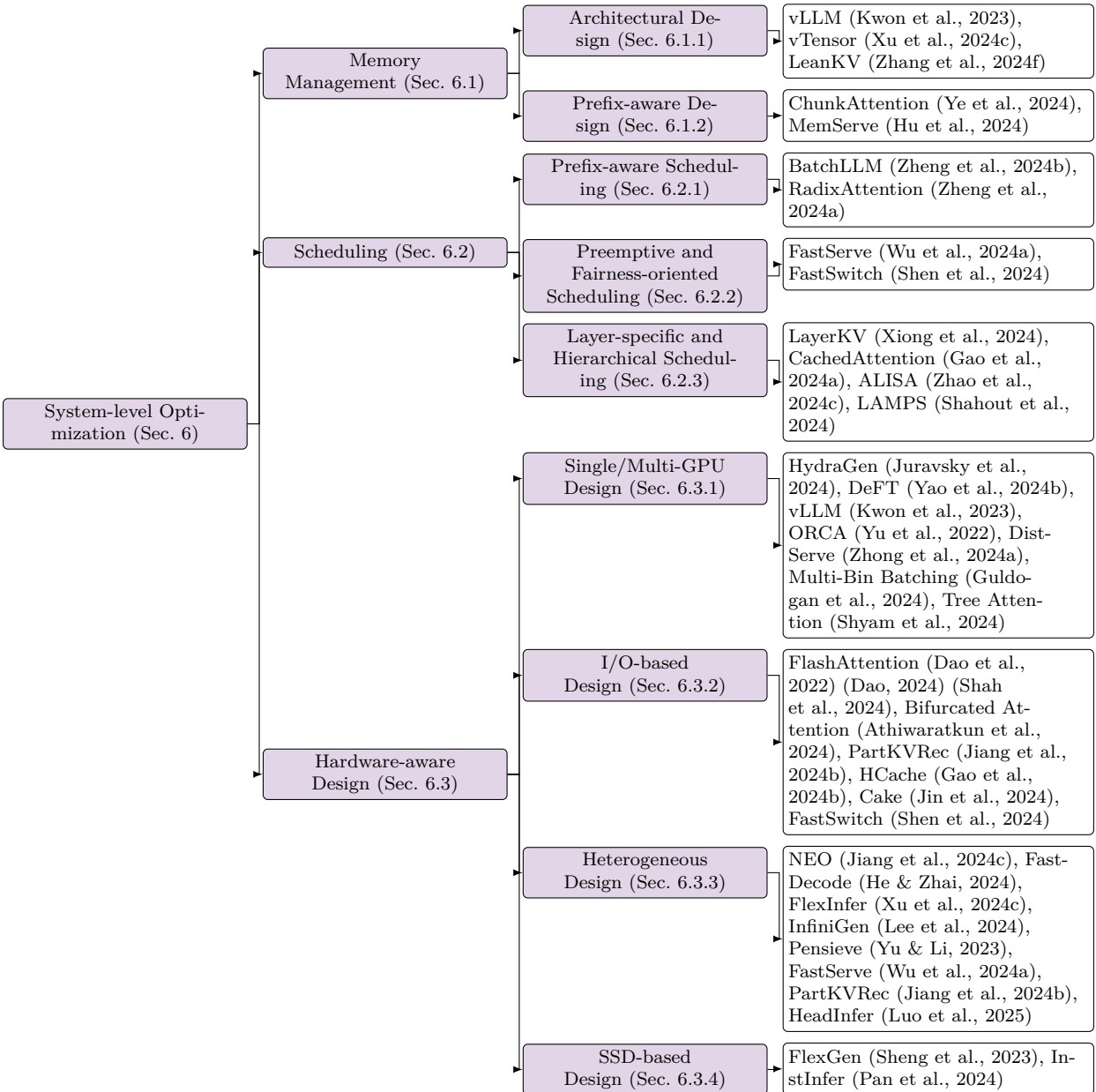

Figure 7: Taxonomy of the System-level Optimization for KV Cache Management.

block management through block tables that track mapping relationships and fill states. This memory management approach enables efficient memory utilization both within and across requests, demonstrating how classical OS memory management principles can be effectively adapted for LLM inference optimization.

This approach is further enhanced by vTensor (Xu et al., 2024c), which introduces a virtual memory abstraction that decouples computation from defragmentation through three key components: the vTensor Scheduler which generates memory management policies based on meta information, the vTensor Operation which translates these policies into CUDA VMM operations, and the vTensor Pool which maintains virtual tensor mappings. VTS processes instructions and creates policies based on memory state tracking, while VTO executes these policies through asynchronous GPU operations. VTP completes the cycle by managing virtual tensor storage and updating meta information for subsequent memory operations.

Table 10: Comparison of Memory Management Techniques for KV Cache Optimization.

| Method | Paged Memory | Virtual Memory | Dynamic Sparsity | Prefix Sharing | Distributed Memory |
|---|:---:|:---:|:---:|:---:|:---:|
| **vLLM**  (Kwon et al., 2023) | ✓ | ✓ | | | |
| **vTensor**  (Xu et al., 2024c) | | ✓ | | | |
| **LeanKV**  (Zhang et al., 2024f) | ✓ | | ✓ | | |
| **ChunkAttention**  (Ye et al., 2024) | | | | ✓ | |
| **MemServe**  (Hu et al., 2024) | | | | ✓ | ✓ |

LeanKV (Zhang et al., 2024f) combines unified paging with heterogeneous quantization and dynamic sparsity mechanisms. It implements Hetero-KV quantization to store keys and values at different precisions, complemented by a per-head dynamic sparsity mechanism that adapts memory allocation based on token importance across different attention heads and requests. To efficiently execute these strategies, LeanKV (Zhang et al., 2024f) introduces an advanced on-GPU memory management system featuring three key components: unified paging for flexible memory organization, a circular free page list for efficient coordination, and a bidirectional page table for minimal metadata overhead.

### 6.1.2 Prefix-aware Design

Some latest works emphasize optimizing data organization structures through prefix-aware designs. ChunkAttention (Ye et al., 2024) restructures KV cache management by organizing chunks within a prefix tree structure, enabling runtime detection and sharing of common prefixes. It breaks down traditional monolithic KV cache tensors into smaller, manageable chunks organized within a prefix tree structure, enabling efficient runtime detection and sharing of common prefixes across multiple requests. This architectural design brings two significant memory management benefits: efficient KV cache deduplication through prefix tree-based organization, and improved data locality through a two-phase partition algorithm for self-attention computation. By enabling dynamic identification and sharing of common prompt prefixes across multiple requests, ChunkAttention (Ye et al., 2024) optimizes both memory utilization and computational efficiency, demonstrating how intelligent chunking and prefix-aware cache management can significantly enhance LLM serving efficiency.

MemServe (Hu et al., 2024) extends this concept to distributed settings with its MemPool system, which orchestrates both CPU DRAM and GPU HBM resources across serving instances, managing active and historical KV caches through a comprehensive set of distributed memory pool APIs. It presents a prompt token-based indexing layer for historical KV cache retrieval, cross-instance data exchange mechanisms that abstract away hardware heterogeneity, and a global scheduler implementing a prompt tree-based locality-aware policy for enhanced cache reuse, collectively resulting in significant improvements in job completion time and time-to-first-token performance.

These approaches often complement each other, suggesting potential benefits in combining multiple strategies. For instance, LeanKV (Zhang et al., 2024f)'s integration of compression with page-based management and MemServe (Hu et al., 2024)'s combination of distributed memory management with prefix-aware caching demonstrate the effectiveness of hybrid approaches. The diversity of these solutions reflects both the complexity of KV cache management and the rich opportunity space for continued innovation in optimizing LLM inference systems. Tab.10 provides a comparison of various memory management techniques for KV Cache, highlighting key features such as paged memory, virtual memory, dynamic sparsity, prefix sharing, and distributed memory.

### 6.1.3 Summary and Future Directions

The exploration of memory management strategies for KV caches in large language model inference reveals a promising landscape of innovations that enhance memory efficiency and overall system performance. Architectural advancements, such as those seen in vLLM (Kwon et al., 2023) and LeanKV (Zhang et al., 2024f),

adapt traditional memory management principles for modern AI applications by incorporating paging and virtual memory concepts for dynamic allocation. Prefix-aware designs like ChunkAttention (Ye et al., 2024) and MemServe (Hu et al., 2024) optimize data organization, enabling the detection and sharing of common prefixes, which reduces redundancy and speeds up inference.

Future memory management work should prioritize adaptive hierarchies, novel compression techniques, intelligent prefetching, and hardware-aware optimizations utilizing new memory technologies, alongside efficient distributed cache coherence. Exploring machine learning for predictive allocation, specialized data structures, and heterogeneous memory systems is also vital for enhancing LLM inference scalability and efficiency.

## 6.2 Scheduling

Based on these scheduling-oriented works, we can categorize KV cache scheduling optimizations into three main approaches: 1) prefix-aware scheduling strategies, represented by BatchLLM (Zheng et al., 2024b) and RadixAttention (Zheng et al., 2024a); 2) preemptive and fairness-oriented scheduling, exemplified by FastServe (Wu et al., 2024a) and FastSwitch (Shen et al., 2024); 3) layer-specific and hierarchical scheduling approaches, demonstrated by LayerKV (Xiong et al., 2024), CachedAttention (Gao et al., 2024a), and ALISA (Zhao et al., 2024c). These approaches collectively address different aspects of scheduling optimization, from memory efficiency to fairness and latency reduction, while specialized solutions like LAMPS (Shahout et al., 2024) extend these concepts to specific use cases such as API-augmented LLM requests, demonstrating the rich design space in KV cache scheduling optimization.

### 6.2.1 Prefix-aware Scheduling

Unlike traditional LRU-based cache management systems where shared KV contexts might be prematurely evicted or unnecessarily extended in memory, BatchLLM (Zheng et al., 2024b) implements explicit global prefix identification and coordinated scheduling of requests sharing common KV cache content. It schedules requests at the granularity of prefix-sharing groups, ensuring optimal KV cache reuse while minimizing cache lifetime - requests with identical prefixes are deliberately scheduled together to maximize KV cache sharing efficiency. This scheduling approach is complemented by a dynamic programming algorithm that optimizes first-level prefix patterns, enabling more efficient KV cache management and reducing scheduling overhead.

RadixAttention (Zheng et al., 2024a) builds around a radix tree structure, replacing traditional FCFS scheduling with an intelligent cache-aware approach that prioritizes requests based on matched prefix lengths. It implements dynamic memory management where cached tokens and running requests share the same memory pool, controlled by an LRU eviction policy that strategically removes leaf nodes while preserving valuable ancestor prefixes. This is complemented by a reference counting mechanism that prevents eviction of actively used cache entries during continuous batching while enabling efficient memory reclamation when nodes become unused.

### 6.2.2 Preemptive and Fairness-oriented scheduling

FastServe (Wu et al., 2024a) implements a proactive KV cache management strategy that coordinates cache movement between GPU and host memory, overlapping data transmission with computation to minimize latency impact. This is integrated with a skip-join Multi-Level Feedback Queue scheduler that makes KV cache scheduling decisions based on input length information, allowing jobs to enter appropriate priority queues directly while avoiding unnecessary demotions through higher-priority queues. By combining token-level preemption with sophisticated KV cache management and intelligent queue placement, FastServe (Wu et al., 2024a) achieves significant performance improvements over traditional run-to-completion systems like vLLM (Kwon et al., 2023).

FastSwitch (Shen et al., 2024) introduces a fairness-oriented KV cache scheduling system that addresses the overhead challenges of preemptive scheduling in LLM serving. There are three key mechanisms: enhancing I/O utilization through intelligent cache movement scheduling, minimizing GPU idle time during context switches, and eliminating redundant I/O operations in multi-turn conversations. Unlike traditional block-based KV cache memory policies that prioritize memory efficiency at the cost of fragmentation and gran-

Table 11: Comparison of Scheduling Approaches for KV Cache Optimization.

| Method | Prefix-aware | Preemptive | Fairness-oriented | Layer-specific | Hierarchical | Dynamic |
|---|---|---|---|---|---|---|
| **BatchLLM** (Zheng et al., 2024b) | ✓ | | | | | |
| **RadixAttention** (Zheng et al., 2024a) | ✓ | | | | | ✓ |
| **FastServe** (Wu et al., 2024a) | | ✓ | ✓ | | | |
| **FastSwitch** (Shen et al., 2024) | | ✓ | ✓ | | | |
| **LayerKV** (Xiong et al., 2024) | | | | ✓ | | |
| **CachedAttention** (Gao et al., 2024a) | | | | ✓ | ✓ | |
| **ALISA** (Zhao et al., 2024c) | | | | ✓ | | ✓ |
| **LAMPS** (Shahout et al., 2024) | | | | | ✓ | ✓ |

ularity limitations, FastSwitch (Shen et al., 2024) implements a balanced approach that maintains efficient memory usage while facilitating smoother context switching. This integrated scheduling approach enables dynamic priority adjustments for fairness while minimizing the performance impact of context switches.

### 6.2.3 Layer-specific and Hierarchical Scheduling

LayerKV (Xiong et al., 2024) introduces a novel layer-wise KV cache scheduling approach to address the growing TTFT (Time to First Token) latency challenges in large-context LLM serving. The contribution lies in its fine-grained, layer-specific KV cache block allocation and management strategy, which departs from traditional monolithic cache management approaches. By implementing layer-wise KV block scheduling and offloading mechanisms, LayerKV (Xiong et al., 2024) enables more efficient memory utilization and reduces queuing delays that typically occur when large context windows compete for limited GPU KV cache blocks. It is complemented by an SLO-aware scheduler that optimizes cache allocation decisions based on service level objectives, allowing for dynamic management of memory resources across model layers.

CachedAttention (Gao et al., 2024a) introduces a hierarchical scheduling approach consisting of three-tier strategies: layer-wise pre-loading coordinates KV cache movement across storage hierarchies using scheduler-aware fetching and eviction policies, asynchronous saving overlaps I/O operations with GPU computation, and intelligent cache placement decisions are made based on scheduler hints to ensure frequently accessed KV caches reside in faster memory tiers. It also presents a novel positional encoding decoupling mechanism that prevents KV cache invalidation during context window overflow through effective truncation strategies.

ALISA (Zhao et al., 2024c) introduces a dual-level KV cache scheduling framework that combines algorithmic sparsity with system-level optimization. At the algorithm level, the Sparse Window Attention mechanism identifies and prioritizes the most important tokens for attention computation, creating a mixture of global dynamic and local static sparse patterns that significantly reduce KV cache memory requirements. At the system-level, its three-phase token-level dynamic scheduler that manages KV tensor allocation and optimizes the trade-off between caching and recomputation. The scheduler makes dynamic decisions about which tokens to cache in GPU memory versus recompute, based on their importance and system resource constraints.

LAMPS (Shahout et al., 2024) implements a predictive scheduling mechanism that estimates both pre-API outputs and optimal memory handling strategies during API calls, choosing between preserving, discarding, or swapping KV cache content based on predicted memory waste.

### 6.2.4 Summary and Future Directions

Tab.11 compares scheduling approaches for KV cache optimization based on their support for prefix-awareness, preemptive scheduling, fairness, layer-specific optimizations, hierarchical structures, and

dynamic adaptability. The advancements in scheduling strategies for KV cache management in large language model inference highlight a multifaceted approach to optimizing performance, memory efficiency, and fairness. By categorizing these strategies into prefix-aware, preemptive and fairness-oriented, and layer-specific scheduling, we see diverse methodologies addressing different challenges. For instance, prefix-aware strategies like BatchLLM (Zheng et al., 2024b) and RadixAttention (Zheng et al., 2024a) enhance cache reuse by intelligently grouping requests based on shared prefixes, minimizing cache lifetime and reducing overhead. Meanwhile, preemptive approaches such as FastServe (Wu et al., 2024a) and FastSwitch (Shen et al., 2024) implement proactive management techniques that optimize cache movement and scheduling, significantly improving latency and ensuring fairness during context switching. Layer-specific scheduling methods like LayerKV (Xiong et al., 2024), CachedAttention (Gao et al., 2024a), and ALISA (Zhao et al., 2024c) further refine cache allocation by implementing fine-grained management strategies tailored to the unique demands of different model layers.

Future KV cache scheduling work should focus on adaptive and predictive systems, automated tuning, context-aware architectures, and novel coherence protocols. Integrating reinforcement learning and hardware-software co-design will further enhance LLM inference system robustness, efficiency, and adaptability. Finally, considering LLM serving (Yao et al., 2024a), different scheduling and sharing for multiple users and queries may lead to potential privacy leaks. Therefore, privacy protection techniques for LLM serving in multi-user scenarios, such as differential privacy (Zhao & Chen, 2022; Dong & Yi, 2021; Dong et al., 2023a), are worth further investigation.

### 6.3 Hardware-aware Design

Recent hardware-aware optimizations for KV cache management span several key directions based on different hardware architectures and constraints. Single/Multi-GPU designs focus on optimizing memory access patterns, GPU kernel designs for efficient attention computation, and parallel processing with load balancing. IO-based designs optimize data movement across memory hierarchies through asynchronous I/O and intelligent prefetching mechanisms. Heterogeneous designs orchestrate computation and memory allocation across CPU-GPU tiers. SSD-based solutions have evolved from basic offloading approaches to more sophisticated designs, with InstInfer leveraging computational storage drives (CSDs) to perform in-storage attention computation, effectively bypassing PCIe bandwidth limitations. These approaches demonstrate how hardware-aware designs can significantly improve LLM inference efficiency by carefully considering and exploiting the characteristics of different hardware components and their interconnections.

### 6.3.1 Single/Multi-GPU Design

Based on these works focusing on GPU-oriented designs, we can categorize the approaches into several key strategies for KV cache optimization. First, shared prefix optimization approaches like HydraGen (Juravsky et al., 2024) and DeFT (Yao et al., 2024b) focus on efficient GPU memory utilization through batched prefix computations and tree-structured attention patterns. Rather than maintaining separate KV caches for each sequence with identical prefixes, HydraGen (Juravsky et al., 2024) decomposes attention computation to leverage a single shared KV cache for common prefixes across multiple requests. It enables efficient GPU memory utilization through two mechanisms: batched prefix KV cache access across sequences and separate handling of unique suffix KV caches. For DeFT (Yao et al., 2024b), its core contributions are twofold: KV-Guided Grouping, which optimizes GPU memory access patterns by intelligently managing shared prefix KV caches to minimize redundant global-to-shared memory transfers, and Flattened Tree KV Splitting, which ensures balanced workload distribution across GPU compute units while minimizing computational redundancy.

Second, distributed processing frameworks exemplified by vLLM (Kwon et al., 2023) and ORCA (Yu et al., 2022) optimize multi-GPU scenarios through sophisticated memory management and synchronization mechanisms. vLLM (Kwon et al., 2023) also implements a KV cache manager that coordinates memory allocation across distributed GPU workers in model-parallel deployments, where each GPU handles a subset of attention heads while sharing the same logical-to-physical block mapping. This GPU-aware design enables efficient memory utilization through near-zero fragmentation and flexible KV cache sharing, while supporting Megatron-LM style tensor parallelism where GPUs execute in SPMD fashion with synchronized block-wise

matrix operations. The scheduler broadcasts control messages containing input tokens and block tables to GPU workers, allowing them to independently process their assigned attention heads while maintaining memory coherence through all-reduce operations, effectively eliminating redundant memory management synchronization overhead and maximizing GPU utilization across distributed resources.

ORCA (Yu et al., 2022) distributes model layers across GPUs using both intra-layer and inter-layer parallelism, where each worker process manages multiple GPU-controlling threads and coordinates KV cache access through an Attention KV manager. ORCA's GPU-aware design minimizes CPU-GPU synchronization overhead by separating control message communication from tensor data transfer (via NCCL), allowing each GPU thread to efficiently access KV cache memory using request IDs and token indices.

Third, phase-aware designs like DistServe (Zhong et al., 2024a) separate prefill and decoding phases across GPU resources to optimize their distinct memory access patterns. Novel batching strategies are represented by Multi-Bin Batching (Guldogan et al., 2024), which focuses on length-aware request grouping for improved GPU utilization, while advanced parallel computation frameworks like Tree Attention (Shyam et al., 2024) introduce sophisticated reduction algorithms for efficient attention computation across multiple GPUs. DistServe (Zhong et al., 2024a) recognizes that prefill and decoding phases have distinct KV cache utilization characteristics and memory access patterns: prefill requires intensive computation with growing KV cache sizes for processing input tokens, while decoding maintains a fixed KV cache size for generating output tokens. By physically separating these phases onto different GPUs, DistServe enables optimized GPU memory management and KV cache access patterns specific to each phase, eliminating interference between prefill's bursty memory access patterns and decoding's steady-state KV cache utilization. Multi-Bin Batching (Guldogan et al., 2024) introduces a length-aware batching strategy helps minimize GPU idle time and memory fragmentation that typically occurs when processing requests of varying lengths in the same batch, as it ensures that the KV cache memory allocated for each batch is utilized more uniformly across all requests. Tree Attention (Shyam et al., 2024) implements a tree-based reduction algorithm that fundamentally changes how attention values are computed and aggregated across GPUs, enabling more efficient handling of KV cache data through partial reductions that significantly reduce memory bandwidth requirements and peak memory usage.

These approaches can collectively demonstrate how hardware-aware designs can significantly improve the LLM efficiency by carefully considering GPU architecture characteristics and memory hierarchy constraints.

### 6.3.2 I/O-based Design

Recent I/O-focused optimizations for KV cache management span several key dimensions, targeting different levels of the memory hierarchy. At the GPU level, approaches like FlashAttention (Dao et al., 2022) (Dao, 2024) (Shah et al., 2024) and Bifurcated Attention (Athiwaratkun et al., 2024) optimize data movement between HBM and SRAM through sophisticated tiling strategies and split attention computations, while CPU-GPU data movement optimizations are addressed by systems like PartKVRec (Jiang et al., 2024b), which tackles PCIe bandwidth bottlenecks through hybrid recomputation and transfer strategies, and HCache (Gao et al., 2024b), which optimizes intermediate activation storage and restoration.

FlashAttention (Dao et al., 2022) (Dao, 2024) (Shah et al., 2024) employs a tiling strategy that carefully manages KV cache access patterns, reducing redundant memory operations by keeping frequently accessed portions of the KV cache in fast SRAM while systematically fetching and evicting data blocks to minimize HBM accesses. Bifurcated Attention (Athiwaratkun et al., 2024) presents an I/O-aware approach to optimize KV cache access patterns during shared-context batch decoding by strategically splitting attention computations into two distinct GEMM operations. It specifically targets the memory bandwidth bottleneck in high-batch scenarios with long contexts by minimizing repeated KV cache accesses, maintaining the same computational FLOPs while drastically reducing memory I/O operations. For PartKVRec (Jiang et al., 2024b), its key innovation lies in its hybrid strategy of partial KV cache recomputation on the GPU while simultaneously transferring the remaining cache data from CPU memory, effectively hiding PCIe transfer latency. The implementation employs a sophisticated I/O-aware scheduling system that analyzes input characteristics and hardware capabilities to determine the optimal balance between recomputation and data transfer, dynamically managing KV cache movement to maximize PCIe bandwidth utilization while mini-

mizing GPU idle time. HCache (Gao et al., 2024b) strategically stores and restores intermediate activations instead of complete KV cache states, implementing a bubble-free restoration scheduler that carefully balances computation and I/O operations to maximize bandwidth utilization. A key innovation is its chunk-based storage manager that addresses the I/O pattern mismatch between saving (layer-before-token) and restoration (token-before-layer) operations, optimizing data layout and access patterns to reduce I/O overhead. Cake (Jin et al., 2024) addresses the fundamental I/O bottleneck in loading cached KV states from disk to GPU memory. It introduces a bidirectional parallelized strategy that simultaneously leverages both computational and I/O resources. This hybrid approach dynamically balances between loading cached KV states from storage and computing them on GPUs, adapting automatically to varying system conditions without manual parameter tuning.

Context management optimizations are exemplified by FastSwitch (Shen et al., 2024), which implements efficient context switching mechanisms for multi-user scenarios through granular memory management policies. FastSwitch (Shen et al., 2024) addresses I/O inefficiencies in traditional block-based KV cache approaches by implementing a more granular and continuous memory management policy that minimizes I/O overhead during preemption and context switching.

These approaches demonstrate how careful consideration of I/O patterns and memory hierarchy characteristics can significantly improve LLM inference efficiency by minimizing data movement and maximizing bandwidth utilization across different storage tiers.

### 6.3.3 Heterogeneous Design

Recent heterogeneous computing approaches for KV Cache demonstrate diverse strategies for optimizing CPU-GPU collaboration. Systems like NEO (Jiang et al., 2024c), FastDecode (He & Zhai, 2024) and HeadInfer (Luo et al., 2025) implement strategic workload distribution through CPU offloading of attention computations, while FlexInfer (Xu et al., 2024c) introduces virtual memory abstractions for optimal resource coordination.

NEO (Jiang et al., 2024c) advances heterogeneous computing for LLM inference by implementing strategic CPU offloading of attention computations and KV cache states. Through asymmetric GPU-CPU pipelining and load-aware scheduling, it optimally balances workloads across both computing platforms, enabling larger GPU batch sizes without latency penalties. For FastDecode (He & Zhai, 2024), its key contribution lies in its strategic offloading of memory-bound KV cache operations to distributed CPU resources, leveraging the aggregate memory capacity and computing power of multiple CPU nodes rather than treating CPUs as mere storage devices. By utilizing CPUs for KV cache computations and storage while keeping compute-intensive operations on GPUs, it creates an efficient pipeline that maximizes resource utilization across the heterogeneous infrastructure, enabling larger batch sizes and higher throughput. FlexInfer (Xu et al., 2024c) orchestrates CPU-GPU resource utilization for LLM inference by introducing the virtual memory-based abstraction vTensor. By implementing fine-grained, selective offloading of attention heads' KV cache to CPU RAM while dynamically computing attention outputs, HeadInfer (Luo et al., 2025) achieves remarkable memory efficiency without compromising computational performance. It also supports both dense and sparse attention mechanisms, integrates with pipeline parallelism for larger models. Unlike NEO (Jiang et al., 2024c), which focuses on strategic CPU offloading with asymmetric GPU-CPU pipelining, and FastDecode (He & Zhai, 2024), which distributes KV cache operations across multiple CPU nodes, HeadInfer distinguishes itself through its granular head-wise offloading strategy that eliminates the need to fully store KV cache for any transformer layer on GPU while preserving complete mathematical equivalence without approximation.

Advanced caching and prefetching mechanisms are exemplified by InfiniGen (Lee et al., 2024), which employs speculative prefetching for KV cache entries, and Pensieve (Yu & Li, 2023), which implements multi-tier caching for conversation states. For InfiniGen (Lee et al., 2024), its key innovation lies in its prediction mechanism that operates across the heterogeneous architecture, using partial computation of attention inputs and modified query-key weights to identify and prefetch only the most relevant KV cache entries from CPU memory to GPU. Pensieve (Yu & Li, 2023) introduces a heterogeneous computing architecture specifically designed for multi-turn conversation LLM serving by implementing a sophisticated multi-tier caching

Table 12: Comparison of Hardware-aware Design Approaches for KV Cache Optimization.

| Method | Single/Multi-GPU | I/O-aware | Heterogeneous | SSD-based |
|---|:---:|:---:|:---:|:---:|
| **Bifurcated Attention** (Athiwaratkun et al., 2024) | | ✓ | | |
| **Cake** (Jin et al., 2024) | | | | ✓ |
| **DeFT** (Yao et al., 2024b) | ✓ | | | |
| **DistServe** (Zhong et al., 2024a) | | | ✓ | |
| **FastDecode** (He & Zhai, 2024) | | ✓ | | |
| **FastSwitch** (Shen et al., 2024) | ✓ | | | |
| **FlexGen** (Sheng et al., 2023) | | ✓ | | |
| **FlexInfer** (Xu et al., 2024c) | | | | ✓ |
| **FlashAttention** (Dao et al., 2022) (Dao, 2024) (Shah et al., 2024) | ✓ | | ✓ | |
| **HCache** (Gao et al., 2024b) | | | ✓ | |
| **HydraGen** (Juravsky et al., 2024) | ✓ | | | |
| **InfiniGen** (Lee et al., 2024) | | | ✓ | |
| **InstInfer** (Pan et al., 2024) | | | | |
| **Multi-Bin Batching** (Guldogan et al., 2024) | | | | ✓ |
| **NEO** (Jiang et al., 2024c) | | | ✓ | |
| **ORCA** (Yu et al., 2022) | ✓ | | | |
| **PartKVRec** (Jiang et al., 2024b) | | ✓ | | |
| **Pensieve** (Yu & Li, 2023) | | ✓ | | |
| **Tree Attention** (Shyam et al., 2024) | | ✓ | | |
| **vLLM** (Kwon et al., 2023) | ✓ | | | |

strategy across GPU and CPU resources. This stateful approach manages KV cache data across the heterogeneous memory hierarchy, maintaining conversation history states across multiple hardware tiers rather than recomputing them for each interaction.

Sophisticated scheduling and preemption strategies are demonstrated by FastServe (Wu et al., 2024a), which focuses on token-level preemption and proactive memory management, and PartKVRec (Jiang et al., 2024b), which balances data transfer and recomputation through dynamic scheduling. For FastServe (Wu et al., 2024a), its token-level preemption capability is supported by a sophisticated heterogeneous memory management system that proactively coordinates KV cache data movement between GPU and host memory. It implements a skip-join Multi-Level Feedback Queue scheduler that manages computational resources across the CPU-GPU boundary, optimizing both computation scheduling and data movement. PartKVRec (Jiang et al., 2024b) employs a scheduler that dynamically optimizes the distribution of tasks across the heterogeneous hardware platform, using a profiler to analyze both hardware capabilities and workload characteristics.

These approaches collectively showcase how heterogeneous architectures can be effectively leveraged to overcome single-device limitations while maintaining efficient resource utilization and minimizing communication overhead between CPU and GPU resources.

### 6.3.4 Solid-state Disk (SSD)-based Design

Recent SSD-based approaches for KV cache management demonstrate an evolution in storage utilization strategies, from traditional extension of the memory hierarchy to computational storage innovations. FlexGen (Sheng et al., 2023) introduces an SSD-based approach to KV cache management that extends the memory hierarchy across GPU, CPU memory, and disk storage, optimizing high-throughput LLM inference on resource-constrained hardware through intelligent tensor storage and access pattern optimization determined by linear programming. The system's key innovations include coordinated data placement across all three storage tiers, optimized access patterns to minimize SSD latency impact, aggressive 4-bit compression for both model weights and attention cache, and efficient utilization of SSD storage as a memory hierarchy

extension for KV cache management. InstInfer (Pan et al., 2024) introduces a more revolutionary approach by leveraging computational storage drives (CSDs) to perform attention computations directly within the storage layer, transforming SSDs from passive storage devices into active computational units and utilizing the high internal bandwidth of flash memory channels to bypass traditional PCIe bandwidth limitations.

These approaches demonstrate how storage devices can be effectively integrated into LLM inference systems, either as memory hierarchy extensions or as computational resources, to enable efficient processing of large models and long sequences in resource-constrained environments. Tab.12 compares hardware-aware design approaches for KV cache optimization across four key features: Single/Multi-GPU support, I/O-awareness, heterogeneous computing, and SSD-based design.

### 6.3.5 Summary and Future Directions

Recent advancements in hardware-aware designs for KV cache management emphasize optimizing performance based on specific hardware architectures and constraints, demonstrating significant enhancements in large language model inference efficiency. Approaches like HydraGen (Juravsky et al., 2024) and vLLM (Kwon et al., 2023) in single and multi-GPU designs focus on efficient memory access patterns and load balancing, while I/O-based strategies such as FlashAttention (Dao et al., 2022; Dao, 2024; Shah et al., 2024) and PartKVRec (Jiang et al., 2024b) tackle data movement bottlenecks through intelligent prefetching and scheduling mechanisms. Additionally, heterogeneous designs exemplified by NEO (Jiang et al., 2024c) and FastDecode (He & Zhai, 2024) effectively leverage CPU-GPU collaboration to maximize resource utilization.

Future research will focus on enhancing LLM inference systems through interconnected advancements in architectural design, hybrid systems leveraging computational storage and in-memory processing, adaptive resource allocation algorithms, advanced compression techniques, and intelligent scheduling for heterogeneous computing. These efforts aim to improve performance and scalability across diverse deployments while adapting to new hardware and demands.

## 7 Long-context Text and Multi-modal Benchmarks

In this section, we introduce the text and multi-modal datasets used to evaluate LLM efficiency.

### 7.1 Text Benchmarks

We collect a lot of long-context datasets, such as NumericBench (Li et al., 2025)and LongBench (Bai et al., 2023). We categorize these datasets into various tasks, including question answering, text summarization, text reasoning, text retrieval, text generation, and aggregation. The brief descriptions of each dataset are provided below, and the corresponding statistics are listed in Tab. 13

- **NumericBench** (Li et al., 2025) is a benchmark designed to evaluate the fundamental numerical reasoning capabilities of LLMs, emphasizing their ability to understand the meaning of numbers.
- **RULER** (Hsieh et al., 2024) is a synthetic benchmark designed to evaluate the long-context capabilities of LLMs across diverse task categories, including retrieval, multi-hop tracing, aggregation, and QA.
- **OneRuler** (Kim et al., 2025) is a multilingual benchmark that evaluates long-context language models across 26 languages.
- **L-Eval** (An et al., 2023) is a benchmark for evaluating long-context language models (LCLMs) across 20 diverse tasks,including closed-ended (e.g., reasoning) and open-ended (e.g., summarization) scenarios.
- **M4LE** (Kwan et al., 2023) introduces a new benchmark for LLMs' long-context comprehension, covering 36 datasets across 11 tasks and 12 domains, with input lengths from 1K to 8K words.
- **BAMBOO** (Dong et al., 2023b) evaluates LLMs' long-text understanding across 5 tasks, including QA, hallucination detection, text sorting, language modeling, and code completion.
- **LongBench** (Bai et al., 2023) is a comprehensive bilingual benchmark designed to evaluate the long-context understanding capabilities of large language models.

Table 13: Long-context Text Benchmarks.

| Benchmark | Tasks | | | | | | Language |
|---|---|---|---|---|---|---|---|
| | Q-A | Summarization | Reasoning | Retrieval | Generation | Aggregation | |
| NumericBench (Li et al., 2025) | ✓ | ✓ | ✓ | ✓ | | | EN |
| RULER (Hsieh et al., 2024) | ✓ | | ✓ | ✓ | | ✓ | EN |
| OneRuler (Kim et al., 2025) | | | | ✓ | | ✓ | 26 languages |
| L-Eval (An et al., 2023) | ✓ | ✓ | ✓ | ✓ | ✓ | | EN |
| M4LE (Kwan et al., 2023) | ✓ | ✓ | | ✓ | | | EN/ZH |
| BAMBOO (Dong et al., 2023b) | ✓ | | ✓ | | ✓ | ✓ | EN |
| LongBench (Bai et al., 2023) | ✓ | ✓ | ✓ | ✓ | ✓ | ✓ | EN/ZH |
| SCROLLS (Shaham et al., 2022) | ✓ | ✓ | ✓ | | | | EN |
| ZEROSCROLLS (Shaham et al., 2023) | ✓ | ✓ | | | | ✓ | EN |
| LooGLE (Li et al., 2023a) | ✓ | ✓ | ✓ | ✓ | | | EN |
| LongEval (Li* et al., 2023) | ✓ | ✓ | ✓ | ✓ | ✓ | | EN |
| StreamingEval (Xiao et al., 2024c) | ✓ | | | ✓ | | | EN |

- **SCROLLS** (Shaham et al., 2022) is a benchmark designed to evaluate models' ability to process naturally long texts across diverse domains such as literature, science, and entertainment.

- **ZEROSCROLLS** (Shaham et al., 2023) serves as a benchmark aimed at assessing the zero-shot reasoning abilities of language models over extended texts.

- **LooGLE** (Li et al., 2023a) benchmarks LLMs' long-context reasoning using post-2022 documents, emphasizing reasoning over memorization, though future training may impact zero-shot evaluation.

- **LongEval** (Li* et al., 2023) is a benchmark for assessing LLMs' long-context capabilities, focusing on tasks requiring reasoning over extended text inputs.

- **StreamingEval** (Xiao et al., 2024c) is a benchmark designed to assess the ability of instruction-tuned LLMs to perform question answering in streaming contexts, where input sequences grow continuously.

## 7.2 Multi-modal Benchmark

Multi-modal datasets combine various data types, such as text, images, audio, and video, to capture the complexity of the real world. Detailed discussions are provided for each benchmark, as outlined in Table 14.

- **LLaVA-Bench** (Liu et al., 2023b) is structured around image-ground-truth textual description-question-answer triplets, segmented across COCO and In-The-Wild datasets.

- **MMBench** (Yuan Liu et al., 2023) serves as a bilingual multi-modal benchmark, facilitating a comparative analysis of VLM performance across English and Chinese linguistic contexts.

- **MileBench** (Song et al., 2024) evaluates the multi-modal long-context capabilities of LLMs, including both diagnostic and realistic evaluation sets. It emphasizes long-context and multi-image tasks.

- **MLVU** (Zhou et al., 2024a) is a comprehensive benchmark for evaluating multi-modal LLMs' video comprehension, with longer videos, diverse genres, and varied assessment tasks.

Table 14: Multi-modal Benchmark Tasks. Specifically, for task abbreviation, **Conv**: conversation task; **Desc**: description task; **Reas**: reasoning task; **Perc**: perception task; **Pred**: prediction task; **SUMM**: summary task.

| Benchmark | Tasks | | | | | | | | | Language |
|---|---|---|---|---|---|---|---|---|---|---|
| | Conv | Desc | Reas | Perc | Pred | Count | Retrieval | Order | SUMM | |
| LLaVA-Bench (Liu et al., 2023b) | ✓ | ✓ | ✓ | | | | | | | EN |
| MMBench (Yuan Liu et al., 2023) | | | ✓ | ✓ | | | | | | EN/ZH |
| MileBench (Song et al., 2024) | | | | | ✓ | ✓ | ✓ | | | EN |
| MLVU (Zhou et al., 2024a) | | ✓ | ✓ | | | ✓ | ✓ | ✓ | ✓ | EN |
| LongVideoBench (Wu et al., 2024b) | | | ✓ | | | | ✓ | | | EN |
| Video-MME (Fu et al., 2024a) | | | ✓ | ✓ | | | | | | EN |
| NExT-QA (Xiao et al., 2021) | | ✓ | ✓ | | | | | | | EN |
| MVBench (Li et al., 2023b) | | | ✓ | ✓ | | ✓ | | | | EN |
| MSVD-QA (Xu et al., 2017) | | ✓ | | | | | | | | EN |
| MSRVTT-QA (Xu et al., 2017) | | ✓ | | | | | | | | EN |

- **LongVideoBench** (Wu et al., 2024b) offers a framework aimed at assessing the capacity of large multi-modal models to comprehend lengthy videos with subtitles, extending up to an hour.

- **Video-MME** (Fu et al., 2024a) evaluates large multi-modal models' video analysis using 900 videos across diverse domains, ranging from 11 seconds to 1 hour for broad scenario coverage.

- **NExT-QA** (Xiao et al., 2021) boasts a dataset with 5,440 videos and approximately 52K manually annotated question-answer pairs, sorted into causal, temporal, and descriptive categories.

- **MVBench** (Li et al., 2023b) comprises 200 multiple-choice question-answer (QA) pairs for each of the 20 temporal understanding tasks, amassing a total of 4,000 QA pairs.

- **MSVD-QA** (Xu et al., 2017) is a collection of 1,970 video clips with descriptive captions, initially for video captioning.

- **MSRVTT-QA** (Xu et al., 2017) comprises 10,000 video clips with 20 human-transcribed sentences each, focusing on connecting video content with language descriptions.

## 7.3 Evaluation Metric

In evaluating LLMs, multiple metrics are essential to comprehensively assess performance across various tasks and application scenarios. Below is a list of commonly used evaluation metrics with brief descriptions.

- **Exact Match (EM)** (Rajpurkar et al., 2016) is a strict metric assessing model accuracy by requiring predictions to exactly match the ground truth.

- **Partial Match (PM)** metric evaluates model output similarity by allowing partial credit for overlaps with the reference, unlike strict metrics like Exact Match (EM).

- **Accuracy** is a metric used to evaluate the overall performance of a model by measuring the proportion of correctly predicted instances (both positive and negative) out of the total instances.

- **Recall** measures a model's ability to retrieve all relevant instances, calculated as the ratio of correctly retrieved items to total relevant items.

- **Precision** is a metric used to evaluate the accuracy of a model by measuring the proportion of correctly predicted positive instances out of all predicted positive instances.

- **F1** combines Precision and Recall using their harmonic mean, offering a balanced evaluation by considering both false positives and negatives.

- **BLEU** (Papineni et al., 2002) evaluates machine translation by comparing n-gram overlap with references, penalizing short outputs for fluency.

- **SacreBLEU** (Post, 2018) standardizes BLEU by fixing preprocessing steps for consistent machine translation evaluations.

- **Rouge** (Lin, 2004) and its variants measure the performance of models by calculating the overlap between the model output and the reference answer with unigram(**Rouge-1**), bigram(**Rouge-2**), LCS(**Rouge-L**).

- **METEOR** (Denkowski & Lavie, 2011) (Metric for Evaluation of Translation with Explicit ORdering) is a text evaluation metric designed to assess the quality of machine translation.

- **BERT** (Zhang et al., 2020) metric, often referred to as BERTScore, is a text evaluation metric that uses contextual embeddings from the BERT model to compare similarity between generated and reference texts.

- **Edit Similarity** measures text sequence similarity by the minimum edits needed to transform one sequence into another, derived from edit distance concepts.

- **Pass@k** (Chen et al., 2021) evaluates the performance of a model by measuring the percentage of cases in which at least one of the top $k$ generated outputs contains a correct solution.

- **Exponential Similarity** is a metric that measures the similarity between two items by exponentially weighting their differences, giving more importance to smaller discrepancies.

- **Concordance Index** is a metric used to evaluate the predictive accuracy of models, particularly in survival analysis or ranking tasks.

- **Mean Reciprocal Rank (MRR)** evaluates ranked results in information retrieval by averaging the reciprocal rank of the first relevant item across queries.

- **Relative Score** evaluates multi-modal models in LLaVA-Bench by comparing outputs to a reference model like GPT-4, calculating the percentage ratio based on helpfulness, relevance, accuracy, and detail.

- **M-Avg** (Multiple-Choice Average) is the mean accuracy across all multiple-choice tasks in the MLVU benchmark, based on the proportion of correct answers.

- **G-Avg** (Generation Average) is the mean score of generation tasks in the MLVU benchmark, evaluated on dimensions like accuracy and relevance using GPT-4, with scores from 1 to 5.

- **WUPS** (K et al., 2012) measures semantic similarity between words based on their taxonomy positions, using the least common ancestor.

## 8 Conclusion and Future Work

Advancements in LLMs have driven significant progress in various fields, but their high computational and memory demands during inference pose challenges, especially for long-context and real-time applications. KV cache management offers an effective solution by optimizing memory, reducing redundant computation, and improving performance. This survey reviews KV cache management strategies across token-level, model-level, and system-level optimizations. Token-level optimizations focus on fine-grained control of KV cache through selection, budget allocation, merging, quantization, and low-rank decomposition, enabling efficient resource allocation without altering model architectures. Model-level optimizations leverage architectural innovations, such as attention grouping and non-transformer designs, to enhance the efficiency of KV reuse. System-level optimizations further complement these efforts by employing advanced memory management, scheduling techniques, and hardware-aware designs to optimize resource utilization across diverse computing environments.

Future directions for KV cache management research include several key areas. First, cross-category integration of token-level, model-level, and system-level optimizations presents significant opportunities but remains underexplored due to the complexity of evaluating combinatorial configurations. Second, real-world case studies are essential to understand how KV cache techniques perform in production environments, shedding light on practical trade-offs, domain-specific priorities, and implementation challenges. Third, domain-specific optimizations, such as retaining critical tokens in healthcare or applying structure-aware strategies in legal and scientific applications, can improve efficiency by tailoring techniques to unique requirements. Fourth, privacy and security considerations, including privacy-aware eviction algorithms and secure isolation mechanisms, are critical for protecting sensitive data in multi-tenant environments. Finally, addressing implementation complexity, scalability, and integration with existing frameworks will be vital to ensure the widespread adoption and practical use of KV cache strategies across diverse applications.

## Acknowledegments

We would like to thank the reviewers and editors of TMLR for their constructive comments. Prof. Lei Chen's work is partially supported by National Key Research and Development Program of China Grant No. 2023YFF0725100, National Science Foundation of China (NSFC) under Grant No. U22B2060, Guangdong-Hong Kong Technology Innovation Joint Funding Scheme Project No. 2024A0505040012, the Hong Kong RGC GRF Project 16213620, RIF Project R6020-19, AOE Project AoE/E-603/18, Theme-based project TRS T41-603/20R, CRF Project C2004-21G, Guangdong Province Science and Technology Plan Project 2023A0505030011, Guangzhou municipality big data intelligence key lab, 2023A03J0012, Hong Kong ITC ITF grants MHX/078/21 and PRP/004/22FX, Zhujiang scholar program 2021JC02X170, Microsoft Research Asia Collaborative Research Grant, HKUST-Webank joint research lab and 2023 HKUST Shenzhen-Hong Kong Collaborative Innovation Institute Green Sustainability Special Fund, from Shui On Xintiandi and the InnoSpace GBA. Prof. Qing Li is supported by the Hong Kong Research Grants Council under General Research Fund (project no. 15200023) and Research Impact Fund (project no. R1015-23). Dr. Haoyang Li is supported by research funds P0052504 and P0053707.

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
