# OpenReview forum: "A Survey on Large Language Model Acceleration based on KV Cache Management"
_TMLR — Accepted by TMLR_

### Review · Reviewer_85cD · 2025-03-11

**Summary Of Contributions:**

This is a broad survey about the topic of KV Cache compression. This is a crucial aspect of efficient LLM inference, and while there are other surveys on the same topic (e.g. https://arxiv.org/pdf/2407.18003) , this work provides a broader overview and considers not only high level methods, but also low level kv cache management and engineering methods.

**Audience:**

Yes

**Broader Impact Concerns:**

I don't have any concerns.

**Claims And Evidence:**

Yes

**Requested Changes:**

See weaknesses.

**Strengths And Weaknesses:**

**Strengths**:
- It was not trivial to design a decent categorization given that there are so many methods, and some of them inevitably overlap. Yet, the way the paper is organized makes a lot of sense.
- The survey is very very exhaustive.
- Fig.2 helps a lot framing the topic and consulting the work, even for readers that might not be from the the field

**Weaknesses**
- In the preliminary section I got the feeling that some parts were written a bit "in a rush".
   - On page 2. "As the computational and memory requirements grow quadratically with sequence length."  -> this sentence is a bit ambiguous. The memory for storing the attention matrix is quadratic but the memory for KV Cache is linear wrt to sequence length. I would clarify this.
   - On page 3. The positional embedding section mentions RoPE very quickly and spends more time on absolute positional encoding that are not used anymore. I wold explain that RoPE is quite different because it is added at each layer and might have impact on kv cache compression.
   - On page 6, section "saved time". I wouldn't call this section "saved time" as what it is actually doing is estimating the saved FLOPs, which of course impact the time. However, the impact depends on many factors, like hardware, and this can be confusing. Maybe a better name for the sections would be computational complexity & memory complexity.

- Figure 2. It looks a bit strange to have the Dataset and Benchmarks in Fig. 2, given that the plot is about KV Cache compression methods. I would regard that section more as a "bonus" rather than placing it on the same level as KV Cache management methods.

- On page 17. “f we store all KV pairs in the memory with full precision, this cache grows exponentially with longer sequences, increasing the memory and bandwidth requirements significantly.” Why does the cache grow exponentially ?

- On page 20. This sentence is not clear: “Then, we introduce the equivalent transformation function-based approaches”

- The citation for FlashAttention v2 and v3 is missing

- Some works for KV Cache compression are missing in the token eviction and quantization section:
     - L2 norm based compression: https://arxiv.org/pdf/2406.11430
- These two works came out after the submission but they are very related:
     - SVD based compression: https://arxiv.org/abs/2503.02812
     - more for keys less for values: https://arxiv.org/abs/2502.15075

- In the datasets and benchmark sections, another popular dataset is ruler: https://github.com/NVIDIA/RULER , that was also recently proposed for multilingual: https://arxiv.org/abs/2503.01996v1 (I believe the extension was published after the submission of the manuscript)

---

> ### Author Response · Authors · 2025-03-29
> **Reply**
>
> Thank you for your valuable detailed suggestions! We have revised our paper according to your comments, which is highlighted in blue in the revised version. Specifically, we have addressed all your **Required Changes (RC)** as follows.
>
> **RC1**: The three unclear parts in introduction and preliminary: (1). Ambiguous sentences on computational and memory requirements in Page 2; (2).The impact of relative positional embedding (such as RoPE)  on KV management in Page 3; (3) The misused term of saved time in Page 6.
>
> **Reply to RC 1**: Thank you for your valuable suggestions.
>
> (1) We have addressed the ambiguous sentences regarding computational and memory requirements on Page 2.
>
> (2) we have expanded the introduction to relative positional embedding methods, such as RoPE in Section 2.2.1. Since relative positional embeddings can significantly influence the effectiveness of KV compression methods, which is an important point often overlooked by other works, thus we have included a future research direction to explore the impact of relative positional embeddings on KV compression methods.
>
> (3) Lastly, we have corrected the terminology on Page 6.
>
>
>
> **RC 2:** Please remove "Dataset and Benchmarks" from Figure 2 due to irrelevance.
>
> **Reply to RC 2:** We have removed "Dataset and Benchmarks" from Figure 2.
>
>
>
> **RC 3:** The errors related to the quadratic increase in KV cache with the length of inputs.
>
> **Reply to RC 3:** Thank you for pointing out this detailed comment. We have corrected this error on Page 17, clarifying that the cache grows linearly with longer sequences.
>
>
>
> **RC 4:** On page 20. This sentence is not clear: “Then, we introduce the equivalent transformation function-based approaches”
>
> **Reply to RC 4:** Thanks for pointing it out. In section 4.4.3, we have added clarification and detailed motivations for the equivalent transformation function-based approaches, which is highlighted in the blue color in our revised version.
>
> **RC 5:** The issue of missing relevant work citations. (1) FlashAttention v2 [1] and v3 [2]; (2) Relevant KV eviction works [3] and quantization papers [4][5]; (3) Dataset papers [6][7]
>
> **Reply to RC 5:** Thank you for pointing this out! We have included these papers in our revised version.
>
> [1] FlashAttention v2: https://arxiv.org/pdf/2307.08691
>
> [2] FlashAttention v3: https://arxiv.org/pdf/2407.08608
>
> [3] L2 norm based compression: https://arxiv.org/pdf/2406.11430
>
> [4] SVD based compression: https://arxiv.org/abs/2503.02812
>
> [5] More for keys less for values: https://arxiv.org/abs/2502.15075
>
> [6] ruler: https://github.com/NVIDIA/RULER
>
> [7] https://arxiv.org/abs/2503.01996v1

---

### Review · Reviewer_6Kho · 2025-03-14

**Summary Of Contributions:**

This survey paper provides a comprehensive overview of Key-Value (KV) cache management strategies for accelerating Large Language Model (LLM) inference. It introduces a taxonomy categorizing techniques into token-level, model-level, and system-level optimizations.

The paper establishes foundational concepts, explaining transformers, auto-regressive generation, and the role of KV caches in reducing redundant computations. It analyzes KV cache operations' time and space complexities and highlights key management challenges.

Token-level optimizations focus on techniques like KV cache selection, budget allocation, merging, quantization, and low-rank decomposition, offering detailed analyses and future directions. Model-level optimizations explore architectural innovations, including attention grouping, architecture adjustments, and non-transformer models like Mamba and RWKV. System-level optimizations address memory management, scheduling, and hardware-aware designs like virtual memory abstractions and heterogeneous computing.

Finally, the paper compiles text and multi-modal datasets for evaluating LLM efficiency, providing a helpful reference for researchers in KV cache optimization.

**Audience:**

Yes

**Claims And Evidence:**

Yes

**Requested Changes:**

## Critical Changes:

- **Unified Benchmark Results**: Include a comprehensive benchmark comparison of representative methods from each category using standard datasets and metrics. This would significantly enhance the practical value of the survey by helping readers identify which approaches are most effective for specific scenarios. For example, in section 5.1, efficiency analysis, such as theoretical computation complexity for sharing or group, given notation in section 2, and summary of empirical results. One suggestion is that section 7 is largely unnecessary; an alternative is to group the evaluation datasets into long or short contexts and summarize the main KV cache optimization results, highlighting tradeoffs (e.g., more compression, more task performance loss).

- **Expanded Critical Analysis**: Strengthen the comparative analysis between different approaches by more explicitly discussing their limitations, trade-offs, and scenarios where they underperform. This would provide a more balanced view of the field. For example, budget allocation techniques seem similar to cache selection, any core differences? e.g., PyramidKV, PyramidInfer, and DynamicKV also select KV caches and maybe merge into a single section. For cross-layer merging, the section is mostly ideas based discussion, any insights into why they work or not, and any empirical summary of findings, especially for KVSharer?

- **Cross-Category Integration**: Add a dedicated section discussing how techniques from different categories (token-level, model-level, system-level) can be integrated for multiplicative benefits. This is important for practical implementations that often combine multiple approaches.

- **Performance vs. Accuracy Trade-offs**: Enhance the discussion about how different KV cache optimization techniques impact model performance (speed, memory) versus accuracy, as this is a critical consideration for practitioners. For example, any tradeoff summary between dynamic and static cache selection? which scenarios were static or dynamic cache selections are better? Any detailed analysis of dynamic selection overhead or end-to-end efficiency gains? For head-wise budge allocation, are there speed ups, or require special software runtime support for head wise compression?

## Strengthening Changes:

- **Summarizing insights**: It would be better to summarize insights/findings for the KV cache optimizations in the introduction section.

- **Case Studies**: If there are any, include case studies of KV cache management in production systems to bridge the gap between research and practice.

- **Quantitative Comparison Framework**: Propose a standardized framework for evaluating and comparing KV cache management techniques, which would help future research in this area.

- **Ethical and Privacy Implications**: Strengthen the discussion on privacy and security considerations related to KV cache management, particularly for multi-user scenarios.

- **Visualization of Techniques**: The comparison tables are already a good visualization. However, adding more visualization diagrams or figures to illustrate the key concepts and differences between various KV cache management approaches would make the paper more accessible.

- **Implementation Complexity**: Provide more discussion on the implementation complexity and deployment challenges of different techniques, as this impacts their practical adoption.

- **Domain-Specific Optimizations**: Include a section on how KV cache management can be specialized for specific domains (e.g., healthcare, finance, legal) where context requirements may differ.

- **Open Source Implementations**: Compile links to open-source implementations of key techniques, which would be valuable for researchers and practitioners looking to build upon existing work.

- **Expanded Multi-Modal Discussion**: Given the growing importance of multi-modal models, expand the discussion on KV cache management for multi-modal contexts where different modalities may have different caching requirements.

- **Writing Changes**: Section 2.1 can be compressed a lot, authors need to briefly discuss keys and values and how they are used during inference, and minimal context for KV cache is good.

**Strengths And Weaknesses:**

## Strengths:

- **Comprehensive and Timely Coverage**: The paper provides an extensive review of KV cache management techniques across multiple dimensions (token-level, model-level, system-level). It also covers many recent works, making it a timely survey of the rapidly evolving field.

- **Well-Structured Taxonomy**: The hierarchical categorization of methods creates a clear framework for understanding the relationships between different approaches, helping readers navigate the landscape of KV cache optimization.

- **Comparative Analysis**: The paper presents detailed technical descriptions of each method, including mathematical formulations, algorithmic ideas, and implementation considerations.
Within each category, the authors provide comparative tables and analyses that highlight the similarities, differences, and trade-offs between different techniques.

- **Future Research Directions**: Each section concludes with thoughtful discussions on research gaps and potential directions for future work, providing valuable guidance for researchers in the field.


## Weaknesses:

- **Limited Critical Analysis**: While the paper catalogs many techniques, it provides relatively limited critical analysis comparing the effectiveness of different approaches under various conditions or workloads. For example, how the token-level KV cache optimizations behave for prefill and decoding phrases.

- **Insufficient Performance Benchmarks**: The paper lacks a unified benchmark comparison of different KV cache management techniques, making it difficult to identify which approaches perform best in which scenarios. The survey could also better highlight the gap between theoretical proposals and practical implementations, especially identifying which methods ae suitable for real-world deployment (e.g, some methods are not batching friendly and not able to deploy at scale).

- **Limited Discussion on Hybrid Approaches**: There could be more exploration of how different optimization levels (token, model, system) can be combined synergistically. Is there any work that co-optimizes the KV cache from token-model or model-system perspectives?

---

> ### Author Response · Authors · 2025-03-29
>
> Thanks for your valuable comments. We have revised our paper according to your comments, which is highlighted in blue in the revised version. We summarize your points from three aspects (1) writing clarification, (2) extensive discussions, (3) future directions.
>
> **Writing Clarification for Changes:** (1) add more visualization diagrams or figures for illustrations (2) summarize the insight of  techniques. (3) compress the unnecessary text. (4) compile links to open-source implementations of key techniques.
>
> **Reply to Writing Clarification Changes:** Thanks for your suggestions. We have revised our paper to enhance clarity and readability as follows.
>
> (1) First, we have added additional figures to cover key aspects such as sparse KV introduction in Figure 4, quantization techniques in Figure 5, various QKV methods in Figure 8, and system components for optimization in Figure 9.
>
> (2) Second, we have summarized the key insights in the introduction and in each corresponding section to provide a more concise and reader-friendly presentation.
>
> (3) Third, we have compressed the preliminaries and dataset descriptions by replacing detailed listings of datasets with an overview of existing benchmarks and citations to the relevant papers.
>
> (4) Lastly, we have collected and consolidated the source code for existing methods (if publicly available) into a GitHub repository for public use. However, to comply with the review process and ensure anonymity, we have not included the URL for the repository in the paper.
>
>
>
> **Extensive Discussions:** (1) Cross-Category Integration (token-level, model-level, system-level) can be explored for achieving multiplicative benefits. (2) Privacy and security considerations related to KV cache management require further investigation. (3) Domain-Specific Optimizations, such as those in healthcare, could be tailored for enhanced performance. (4) Implementation complexity and deployment challenges of various techniques need to be addressed comprehensively. (5) Case discussion on KV cache managements.
>
> **Reply to Extensive Discussions:**  Thanks for providing so comprehensive comments. We have revised paper to add extensive discussion in Section 6.4.
>
>
>
> **Future Directions**: Experiments: A quantitative comparison framework with case studies is needed to evaluate existing KV cache management approaches across various metrics, such as accuracy, memory usage, and speed.
>
> **Reply to Future Directions:**  we agree that a quantitative comparison framework for KV cache management is important, particularly given the hundreds of existing KV cache models. However, due to the limited GPU resources available to our group, we consider developing this framework as a future research direction.

---

### Review · Reviewer_2duU · 2025-03-16

**Summary Of Contributions:**

This paper covers a wide range of KV cache management techniques for LLMs, from low-rank compression to system-level optimizations. The KV cache becomes a major bottleneck during LLM inference, and this paper highlights a range of works that attempt to tackle this.

**Audience:**

Yes

**Claims And Evidence:**

Yes

**Requested Changes:**

1. In Section 4.5.1: "Firstly, ECKVH Yu et al. (2024), EigenAttention Saxena et al. (2024), and ZDC Zhang & Shen (2024) shows
that KV caches have a low-rank property, where a small number of top singular values retain most of the
information. Using Singular Value Decomposition (SVD), the method compresses KV caches by grouping
heads, applying SVD, and retaining top singular values, effectively reducing the number of KV heads with
minimal error." The second sentence is ambiguous and does not describe what Eigen Attention does. Please edit this to specify which method the authors are discussing.

2. There are repeated references (in Introduction) and keywords (eg, "Equation" in 2.2.2) throughout the paper.

3. Please include the following works in the relevant sections:

[1] P. Singhania et al., "Loki: Low-rank Keys for Efficient Sparse Attention", arXiv (2024).

[2] C. Luo et al., "HEADINFER: Memory-Efficient LLM Inference by Head-wise Offloading", arXiv (2025).

[3] Saxena et al., "ResQ: Mixed-Precision Quantization of Large Language Models with Low-Rank Residuals", arXiv (2024).

4. Section 7 on datasets is not needed in this review and makes the paper unnecessarily long. In my opinion, it should be removed and/or summarized by citing related review works on LLM evaluation benchmarks/datasets.

**Strengths And Weaknesses:**

Strengths:

1. This paper is organized well, with figures explaining each section's taxonomy.

2. The authors outline detailed future directions and provide a succinct summary of existing works.

Weaknesses:

1. The paper is unnecessarily long, with a few sections that are not directly related to the KV cache.

---

> ### Author Response · Authors · 2025-03-29
> **Reply**
>
> Thank you for your valuable detailed suggestions. We have revised our paper according to your comments, which is highlighted in blue in the revised version. Specifically, we have addressed all your **Required Changes (RC)** as follows.
>
> **RC 1**: In Section 4.5.1: "Firstly, ECKVH Yu et al. (2024), EigenAttention Saxena et al. (2024), and ZDC Zhang & Shen (2024) shows that KV caches have a low-rank property, where a small number of top singular values retain most of the information. Using Singular Value Decomposition (SVD), the method compresses KV caches by grouping heads, applying SVD, and retaining top singular values, effectively reducing the number of KV heads with minimal error." The second sentence is ambiguous and does not describe what Eigen Attention does. Please edit this to specify which method the authors are discussing.
>
> **Reply to RC1:** thanks for your valuable comments. We have revised this sentence more clearly in our revision paper in Section 4.5.1.
>
> **RC 2**: There are repeated references (in Introduction) and keywords (eg, "Equation" in 2.2.2) throughout the paper.
>
> **Reply to RC2**: We have fixed these typos and errors.
>
> **RC 3:** Please include the following works in the relevant sections
>
> **Reply to RC3:** Thanks for your pointing it. We have added these relevant papers to our survey. Specifically, we included Loki [1] in Section 4.1.3, HeadInfer [2] in Section 6.3.3, and ResQ [3] in Section 4.4.2.
>
> [1] P. Singhania et al., "Loki: Low-rank Keys for Efficient Sparse Attention", arXiv (2024).
>
> [2] C. Luo et al., "HEADINFER: Memory-Efficient LLM Inference by Head-wise Offloading", arXiv (2025).
>
> [3] Saxena et al., "ResQ: Mixed-Precision Quantization of Large Language Models with Low-Rank Residuals", arXiv (2024).
>
> **RC 4:** Section 7 on datasets is not needed in this review and makes the paper unnecessarily long. In my opinion, it should be removed and/or summarized by citing related review works on LLM evaluation benchmarks/datasets.
>
> **Reply to RC4:** Thank you for your valuable suggestions. We have revised Section 7 on datasets by shortening it. Instead of listing all the datasets, we now provide a summary of existing benchmarks and cite the relevant papers associated with these benchmarks.

---

### Decision · Action_Editor_ZRE6 · 2025-04-07

**Recommendation:** Accept with minor revision

**Comment:**

The review process was relatively short, and all comments from the reviewers were included in the new revision. I personally agree that the topic is very interesting, with lots of new research coming out every month. As such, a highly comprehensive survey like this one can be valuable. However, I have some concerns that should be addressed:

1. Please provide a very strong proofreading of the entire paper, as there are several typos and errors, e.g., "by focusing on fine-grained the careful selection", "While KV cache selection (Sec. 4.1) focuses on prioritizing and storing only the most relevant tokens." (should be a comma), "KV cache Architecture alterations (Sec. 5.2" (missing closing bracket), etc.
2. The paper is unnecessarily long, with many repeated parts (e.g., all the "Summary and Future Directions" subsections share similar text), many parts that are only tangentially related to KV cache management, long bullet point lists (e.g., the dataset one). I would suggest to shorten multiple sections as much as possible to make for a clearer, more concise survey. Combining similar subsections is also a valid strategy.

**Audience:**

KV cache management is a core problem in deployment of LLMs (and generally multimodal transformers). Thus, the paper has a potentially very large audience.

**Claims And Evidence:**

The paper is a survey on KV cache management, organized across three broad axes: token-level management, model-level management, and system-level management. The survey overlaps with a lot of similar papers, but the reviewers are all convinced it provides a valuable contribution as (a) the field is evolving quickly, (b) the organization is valid, and (c) the survey is very comprehensive. The paper does not provide empirical benchmarks or recommendations, thus limiting its practical usefulness.

---

> ### Author Response · Authors · 2025-05-09
>
> Dear Editor,
>
> Thanks for your thoughtful review and constructive feedback. We greatly appreciate the time you took to carefully review our work and provide detailed suggestions for improvement. We have revised the manuscript to address your concerns as follows:
>
> - **Typos and Errors:** We conducted a thorough proofreading of the manuscript to correct all typos and errors.
>
> - **Length and Redundancy:** We removed redundant sections, such as similar discussions, and merged overlapping content. In the current version, the manuscript has been reduced by nearly ten pages of content (excluding references) compared to the original.
>
> The updated version has been uploaded for your review. Thank you again for your valuable input.
>
> Sincerely,
>
> Authors

---

> > ### Comment · Action_Editor_ZRE6 · 2025-05-14
> >
> > Dear authors,
> >
> > Please ensure that the camera ready version complies with the guidelines, in particular "*all author information inserted in the manuscript as well as the link to the OpenReview page for the submission*". Proofreading was not performed adequately, as there are still typos (e.g., Liu et al. is cited without brackets in the first paragraph).

---

> > > ### Author Response · Authors · 2025-05-17
> > >
> > > Dear Editor,
> > >
> > > Thank you for your valuable feedback and guidelines.
> > >
> > > In the updated camera-ready version, we have ensured the following:
> > >
> > > * All author information has been properly included in the manuscript.
> > > * The link to the OpenReview page for the submission has been added.
> > > *  We have thoroughly proofread the manuscript and adjusted the citation format throughout the entire text.
> > >
> > > Thank you once again for your time and effort.
> > >
> > > Best regards,
> > >
> > > Authors